# Domain wall motion-driven magnetic convolutional accelerator

Bingqian Dai [1,4] ✉, Tianyi Wang [1,4], Albert Lee [1,4], Shijie Xu [1,4], Chin-Chung Chen[1,2], Kin Wong [1], Dingyi Li[1], Malcolm Jackson[1], Yang Cheng [1], Puyang Huang[1], Yaochen Li[1], Chao Yun[3], Qingyuan Shu[1], Haoran He[1], Lixuan Tai [1], Hanshen Huang[1], Tien-Kan Chung[2], Yanglong Hou[3] ✉ & Kang L. Wang [1] ✉

Modern computing powers applications from data analysis to artificial intelligence but now faces limitations. The slowdown of device scaling and the bottleneck between memory and processors motivate architectures that unify computation and data storage. Convolution is a core operation in learning, vision, and signal processing, yet its conventional implementation incurs high energy, high latency, and limited scalability. Magnetic systems that host spin textures, such as domain walls, offer dynamic behaviors that enable computation beyond traditional logic. Here we introduce a compute-in-memory platform that performs convolution by sequentially shifting magnetic domains and sensing the resulting signals. Information is written directly into domain patterns, processed through controlled motion, and read electrically, forming a nonvolatile structure suited for convolution tasks. This approach supports applications including Fourier analysis, neural networks, and image processing, achieving $10^3$ to $10^5$ improvements in area, energy, and throughput over existing technologies, marking a concrete advance in spintronic computing.

Modern computing underpins transformative advances in artificial intelligence, data analytics, and signal processing. However, it now faces critical challenges. The continued scaling of CMOS technology has encountered severe limitations, signaling the breakdown of Moore's Law, while the separation of memory and processing in the Von Neumann architecture introduces latency and energy bottlenecks. These issues are especially pronounced in workloads with high data movement and computational intensity, such as convolutional operations.

Convolution serves as a cornerstone of modern computing, enabling tasks such as deep learning, image recognition, and frequency-domain signal analysis by extracting spatial and temporal features across hierarchical representations[1–4]. Conventional hardware architectures accelerate convolution through CPUs, GPUs, and TPUs[5,6], utilizing parallelized algorithms to enhance throughput. Yet, these systems remain constrained by the Boolean logic framework, which struggles to support multi-state and sequential operations efficiently[7]. Moreover, the underlying memory hierarchy—spanning registers, caches, RAM, and nonvolatile storage—introduces delays and energy costs associated with frequent data shuttling[8]. These cumulative inefficiencies motivate the search for generalized computing approaches that move beyond Boolean logic and exploit novel physical substrates for both memory and computation.

To address these limitations, emerging technologies are being explored to perform computation via intrinsic physical phenomena, giving rise to compute-in-memory (CIM) paradigms and architectures based on unconventional systems. Among these, spintronic systems based on magnetic solitons—particularly magnetic domain walls (DWs)

[1]Department of Electrical and Computer Engineering, Physics and Astronomy, and Material Science and Engineering, University of California, Los Angeles, CA, USA. [2]Department of Mechanical Engineering, National Yang Ming Chiao Tung University, Hsinchu, Taiwan. [3]School of Materials, Shenzhen Campus of Sun Yat-sen University, Shenzhen, China. [4]These authors contributed equally: Bingqian Dai, Tianyi Wang, Albert Lee, Shijie Xu. ✉e-mail: bdai@g.ucla.edu; hou@sysu.edu.cn; wang@ee.ucla.edu

and skyrmions–exhibit rich dynamic behaviors such as oscillation, motion, and mutual interactions[9–17]. These phenomena can be directly mapped to computational primitives, opening pathways toward low-power, high-speed, and nonvolatile computing architectures. Pioneering studies have demonstrated the immense potential of spintronic phenomena for computation: coupled nanomagnet oscillators have been shown to emulate neural networks[18], spin-wave amplitudes have been used to encode information[19], and DW motion has been harnessed to perform Boolean logic operations[20]. However, no prior work has demonstrated a platform that enables convolutional acceleration while simultaneously integrating memory and computation through spintronic mechanisms.

In this work, we present the Magnetic Convolutional Accelerator (MCA)–a spintronic hardware platform that leverages magnetic DW motion to perform convolution directly within memory. In the MCA, input data are physically encoded as magnetic domains, and computation is realized via sequential DW motion and anomalous Hall effect (AHE) readout, forming a nonvolatile, physical realization of convolution. This approach exploits the analogies between domain shifting and sliding-window operations, enabling the MCA to execute data processing tasks including short-time Fourier transform (STFT), convolutional neural networks (CNNs), and image processing. The system performance can be further enhanced by selecting alternative material systems. For example, systems with large Dzyaloshinskii–Moriya interaction (DMI) offer exceptional scalability (domain sizes down to sub-10 nm[21]); low-roughness systems provide significantly higher energy efficiency (DW motion energy ~27 aJ[22]); and heavy-metal/ferrimagnet systems enable much higher operation speeds (DW velocities exceeding 1000 m/s[23] and dynamic frequencies reaching the THz regime[24]). Importantly, all these systems retain the key advantage of nonvolatility[25].

While a fully general-purpose computer based on emerging technologies and CIM remains a long-term vision–limited today by incomplete algorithm-hardware codesign, device-level maturity, and CMOS integration barriers–our MCA represents a concrete advancement in this direction. It achieves CIM functionality and is particularly suited to edge computing scenarios, where fast, localized, and energy-efficient data processing is critical. Compared to 28 nm CMOS technology, our MCA platform achieves 3 to 5 orders of magnitude improvements in combined area, throughput, and energy efficiency, highlighting its promise as a scalable and sustainable alternative to conventional architectures.

## Results

### MCA architecture and convolution mechanism

Following the introduction of the MCA, we now detail its hardware structure and operating principles, as illustrated in Fig. 1a. The MCA implements convolutional operations using a streamlined and highly integrable architecture based on serially connected magnetic computing unit cells.

Each unit cell consists of a magnetic domain segment overlapped by AHE readout electrodes (Fig. 1b). The magnetic domains are encoded as regions with opposite magnetization: red rectangles represent magnetic moments aligned along the $-z$ direction, and blue rectangles represent moments along the $+z$ direction. Gold-colored strips and pads define the AHE electrodes, which sense the net out-of-plane magnetization within their coverage area.

The Hall voltage ($V_H$) output of a single unit cell is governed by the relation:

$$V_H = \alpha I_H (2L_D - L_P) W_P \tag{1}$$

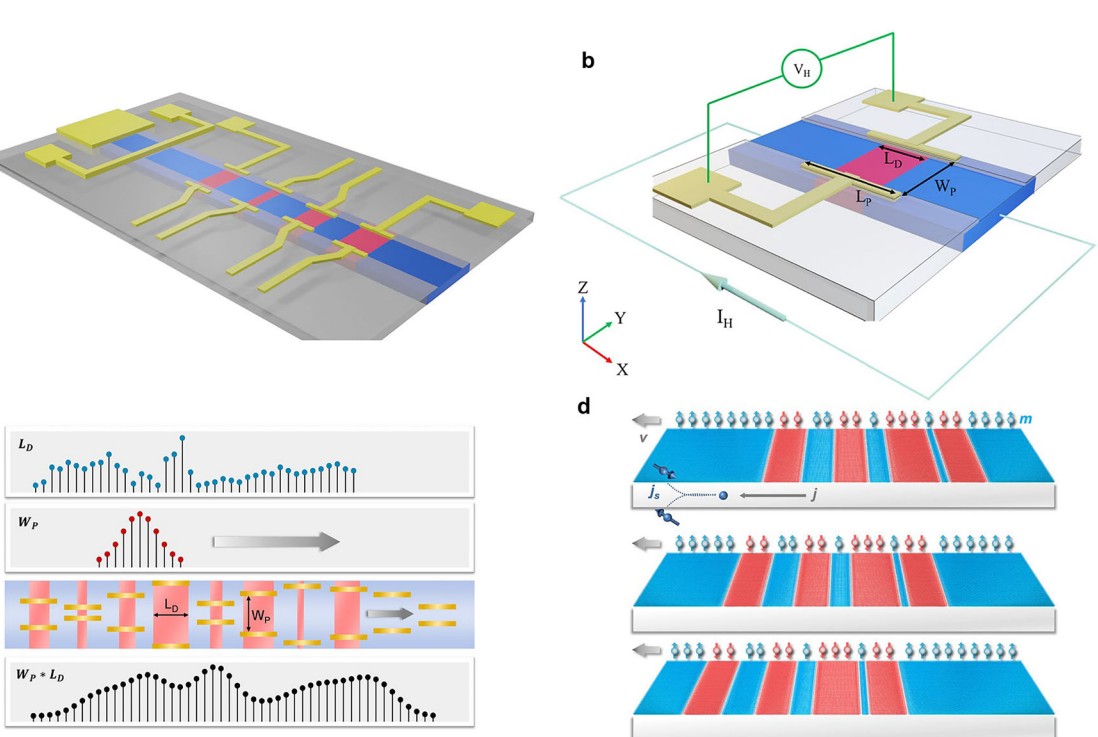

**Fig. 1 | Operating principles of the MCA. a** MCA architecture comprising serially connected magnetic unit cells. **b** Structure of a single unit cell. Red and blue rectangles denote domains with $-z$ and $+z$ magnetization. Gold strips and pads indicate AHE readout electrodes. The domain length is $L_D$, the electrode contact length is $L_P$, and the spacing between Hall-electrode pairs is $W_P$. During readout, a current $I_H$ flows along the conduit and the transverse Hall voltage $V_H$ is measured. **c** Schematic of MCA-based convolution. The input domains (top) and kernel coefficients (second row) are represented by discrete values of $L_D$ and $W_P$. The device-level cartoon (third row) illustrates stationary electrodes and sequential domain shifting. The resulting Hall-voltage sequence (bottom) corresponds to the convolution output. **d** DW shifting driven by SOT. A charge current $j$ generates a transverse spin current $j_s$, which applies a torque on the DWs and translates the domains along the track.

Where $\alpha$ is a device- and material-dependent constant, $I_H$ is the applied current flowing along the +**x** axis of the device, and $L_D$ is the domain length in the **x** direction with the domain magnetization aligned along the $-$**z** direction, $L_P$ is the electrode contact length, and $W_P$ is the lateral spacing between the electrode pairs. This relation captures the combined contributions of domain configuration and electrode geometry to the electrical signal. For a detailed derivation, see "Methods".

In our MCA implementation, we fix $\alpha$, $L_P$, and $I_H$ during operation, enabling a simplified form:

$$V_H = c_1 + c_2 W_P L_D \tag{2}$$

Where $c_1$ and $c_2$ are device-specific constants determined by fabrication. This linear dependence allows the Hall voltage to directly encode the product of domain length and electrode spacing—the two physical quantities mapped to the convolution input and kernel, respectively.

By serially connecting multiple unit cells together, the MCA (Fig. 1a) sums the outputs from each unit cell to form the total Hall voltage:

$$V_{H,TOTAL} = c_1 + c_2 \sum_{n=1}^{N} W_P(n) L_D(n) \tag{3}$$

Where $N$ is the number of unit cells. Critically, in each clock cycle, the domains are shifted by one unit cell length, enabling dynamic convolution across the input data. In each computing cycle, we shift the domains by one unit cell. After $k$ cycles of domain displacement, the total Hall voltage evolves to:

$$V_{H,TOTAL}(k) = c_1 + c_2 \sum_{n=1}^{N} W_P(n) L_D(n-k) \tag{4}$$

This sequential shift-and-readout operation physically implements the convolution operation $(W_P * L_D)[k]$ in real time. The domain lengths $L_D$ encode the input signal, while the electrode spacings $W_P$ define the convolution coefficients.

The working principle, $(W_P * L_D)[k] = \sum_{n=1}^{N} W_P(n) L_D(n-k)$, is illustrated schematically in Fig. 1c. The first row shows the discrete input domains (blue dots) corresponding to varying $L_D$ values. The second row represents the convolutional coefficients (red dots) corresponding to variations in $W_P$, with a gray arrow indicating the shifting direction of the convolution kernel. The device-level cartoon (third row) depicts the magnetic domains, Hall electrodes, and the domain shifting direction (gray arrow). As the domains move sequentially under stationary electrodes, the resulting Hall voltages cumulatively realize the convolution, as shown in the final row.

The domain shifting is enabled by spin-orbit torque (SOT), as depicted in Fig. 1d. A charge current $j$ flowing along the **x**-axis generates a transverse spin current $j_s$ via the spin Hall effect, exerting a torque on the DWs[26]. This spin torque induces coherent DW motion at velocity $v$, governed by the interplay between spin-orbit effects and DMI[9,15]. As a result, the magnetic domains translate uniformly along the conduit, enabling real-time, clocked convolution operations. Further experimental details of domain control are provided in "Methods" and Supplementary Fig. 1.

## Experimental realization of the MCA

The experimental implementation of the MCA can be categorized into three key stages: (1) generation and control of domain lengths ($L_D$) to encode the input signal, (2) DW motion induced by SOT to shift the domains sequentially, and (3) convolution signal readout via the AHE.

Figure 2a schematically illustrates the domain nucleation process. A current-induced Oersted field is applied using an orthogonal conducting or microwave stripline overlaid on the magnetic conduit. Within the magnetic material, gray-shaded regions denote smoothly rotating DW magnetization, transitioning between +**z**-oriented (blue) and $-$**z**-oriented (red) domains. By injecting a short current pulse $I$ into the stripline, a localized Oersted field $H$ is generated, nucleating a +**z**-magnetized domain (blue). The domain length $L_D$ can be finely controlled by tuning the polarity, magnitude, and duration of the current pulse, allowing flexible input encoding. Once nucleated, the domains are shifted collectively by moving their associated DWs, as depicted in Fig. 2b. The DW velocity ($V_{DW}$) is driven by the applied current $j$ and propagates along the conduit in the current direction, enabling synchronized displacement of multiple domains.

A microscope image of a fabricated MCA device is shown in Fig. 2c, highlighting key functional regions for domain generation, shifting, and readout. Details regarding material characterization and device fabrication are provided in Supplementary Figs. 2 and 3.

Figure 2d–h presents experimental demonstrations corresponding to the three operational stages introduced earlier.

(1) Domain generation and control: Fig. 2di illustrates the application of a 200 mA, 1 ms current pulse ($I_{gen}$) into the left Oersted Channel to nucleate a +**z** magnetized domain (blue) within the Domain Channel. As shown in Fig. 2dii, this process produces a domain of ~32 μm length at the left side of the conduit. By varying the polarity and magnitude of $I_{gen}$, different domain lengths ($L_D$) are achieved. Specifically, current pulses of $-80$ mA, $-90$ mA, and $-100$ mA (each 1 ms duration) yield $L_D$ values of 14 μm, 11 μm, and 6 μm, respectively, as displayed in Fig. 2diii. To further refine the domain size control, we performed COMSOL simulations to estimate the spatial distribution and energy consumption of the Oersted field based on device geometry, detailed in Supplementary Fig. 4.

(2) Domain shifting via SOT: Fig. 2e presents the demonstration of sequential domain shifting. The operation is performed by injecting SOT pulses ($I_{shift}$) through the Shifting and Domain Channel (Fig. 2ei). Four magnetic domains, initially spaced at 20 μm intervals (matching the Hall Channel pitch), are sequentially shifted. The corresponding domain-shifting velocity and thermal effects induced by the SOT pulses are experimentally characterized and discussed in Supplementary Note 1 and Note 2, respectively.

As shown in Fig. 2eii, the first domain appears at the right side of the Domain Channel, while the remaining three domains lie outside the field of view. Each SOT pulse displaces the domain sequence by 20 μm toward the left. Figure 2eiii–vi shows the progressive movement of domains into the Hall Channel region. Additional pulses continue the stepwise displacement of the domains, as shown in Fig. 2evii, progressively shifting the domain sequence across the Hall Channel region and completing the full convolution operation.

During the shifting process, small fluctuations in domain length are observed, attributed to domain-wall pinning and imperfect synchronization between neighboring walls. This effect, referred to as the domain-wall synchronization problem, is systematically analyzed in Supplementary Note 3. Complementary micromagnetic simulations further examine the impact of domain-wall pinning in downscaled devices, as shown in Supplementary Note 4.

(3) Convolution signal readout via AHE: When the domains are positioned beneath the Hall electrodes, the convolution signal ($V_{AHE}$) is recorded according to the physical convolution mapping introduced in Eq. (4). Here, the convolution coefficients (kernels) are encoded in the electrode spacings ($W_p$), while the input values are mapped to domain lengths ($L_D$). The granularity of the readout—represented by $\Delta V_{AHE} \sim \Delta W_p \Delta L_D$—is experimentally characterized as a function of both $W_p$ and $L_D$, as shown in Fig. 2f, g.

In Fig. 2f, varying $W_p$ from 2 μm to 20 μm while keeping $L_D = 14$ μm fixed results in a linear increase of $V_{AHE}$ from approximately 0.9 mV to 4.8 mV. Conversely, in Fig. 2g, $L_D$ is varied from 0 μm to 14 μm with a fixed $W_p = 8$ μm, yielding a linear change in $V_{AHE}$

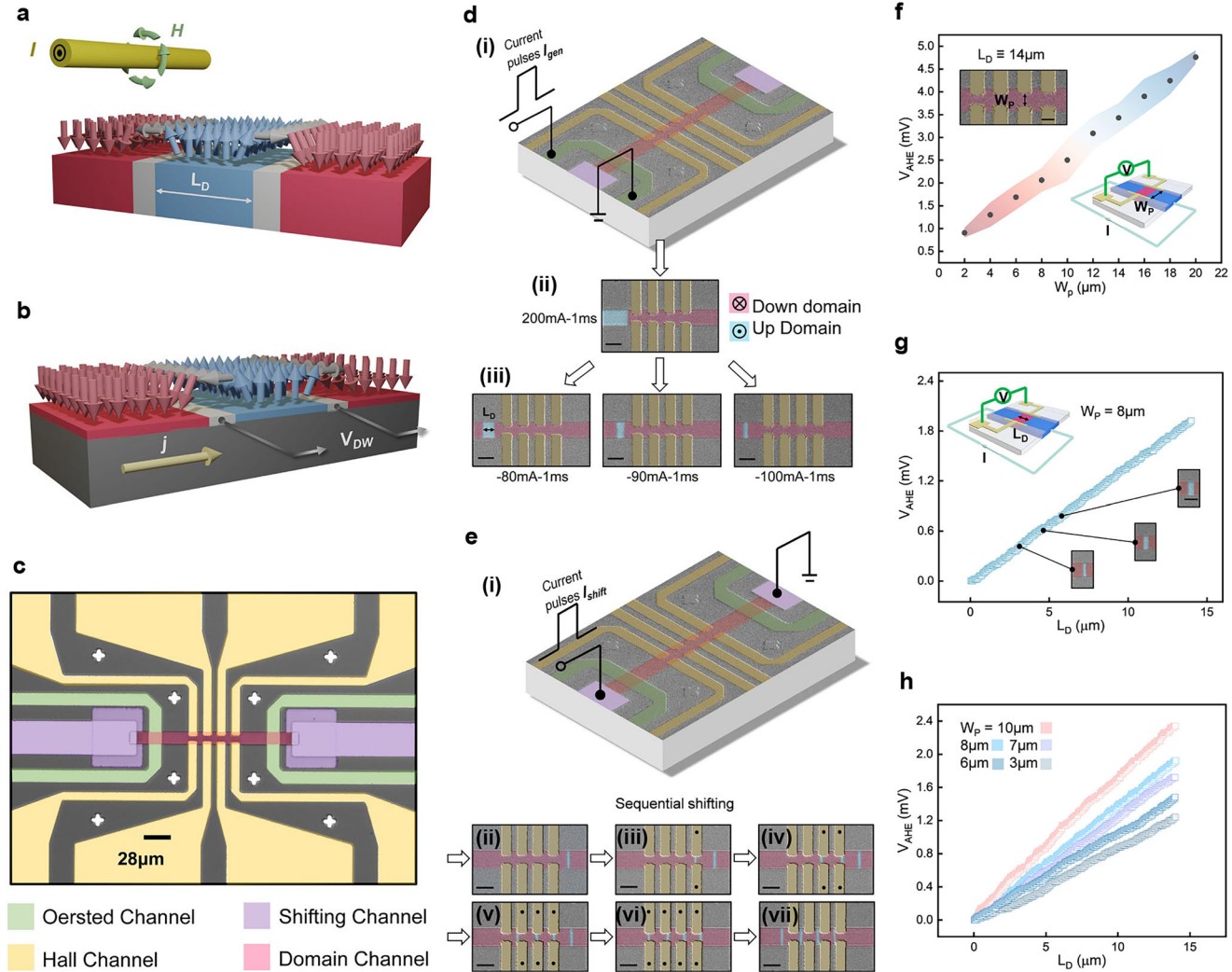

**Fig. 2 | Experimental realization of the MCA. a** Schematic of domain nucleation using current-induced Oersted fields. **b** Schematic of SOT-driven DW shifting. **c** Optical micrograph of a fabricated MCA device. **d** Experimental demonstration of domain nucleation and size control. (i) Setup for generating domains. (ii) Creation of a 32 μm +z domain using a 200 mA, 1 ms pulse. (iii) Domain-length tuning: pulses of −80 mA, −90 mA, and −100 mA (1 ms) yield $L_D$ = 14, 11, and 6 μm, respectively. Scale bar: 20 μm. **e** Experimental demonstration of sequential domain shifting. (i) Setup for applying SOT-driven shifting pulses. (ii) Initial domain position. (iii–vi) Stepwise 20 μm displacement per pulse, shifting domains into the Hall-channels. (vii) Continued pulses translate the full domain sequence across the Hall-channels. Scale bar: 20 μm. **f** AHE-based convolution granularity versus $W_P$ at fixed $L_D$ = 14 μm. Shaded color bands indicate error bars. Insets: device region with varying $W_P$ and measurement setup. Scale bar: 14 μm. **g** AHE-based convolution granularity versus $L_D$ at fixed $W_P$ = 8 μm. Insets: measurement setup and snapshots of domains with different $L_D$. Scale bar: 20 μm. **h** Linear dependence of $V_{AHE}$ on $L_D$ for various electrode spacings $W_p$ (10, 8, 7, 6, and 3 μm).

from 0 to 2 mV. Additional tests of $V_{AHE}$ versus $L_D$ for various $W_p$ values exhibit similar linearity, five examples of which are shown in Fig. 2h. Minor deviations from perfect linearity are attributed to fabrication-induced pinning sites and material imperfections. Further characterization of the current dependence of $V_{AHE}$ is provided in Supplementary Fig. 5 and refer to "Methods" for AHE reading details.

With the successful demonstration of domain generation, shifting, and convolution readout, the complete operational cycle of the MCA is realized. For practical deployment, however, the three functional stages must operate simultaneously and repeatedly with high stability and reproducibility. This requirement raises considerations of device yield and endurance, which directly affect large-scale integration and long-term reliability. To evaluate these aspects, we performed systematic tests of fabrication yield and cycling endurance, with the results summarized in Supplementary Note 5 and Note 6, respectively.

## Algorithm demonstrations on the MCA platform

Building on the physical convolution capability of the MCA, we implement a range of algorithms, including the STFT, CNNs, and image processing tasks. Devices are specifically designed to accommodate these algorithmic requirements. Details of device design and fabrication are provided in "Methods" and Supplementary Fig. 6. In this demonstration, the raw experimental data from Fig. 2f is used as the convolutional coefficients, while linearized input data, extracted from Fig. 2h and presented in Supplementary Fig. 7, is employed as the input signals.

The STFT is a widely utilized signal processing technique for analyzing non-stationary signals whose frequency content evolves over time. It represents the most prevalent form of the discrete Fourier transform (DFT) applied to segmented time-domain signals and serves as a foundational operation in frequency-domain analysis.

As schematically illustrated in Fig. 3a, the STFT process involves segmenting an input signal into overlapping windows, each of which

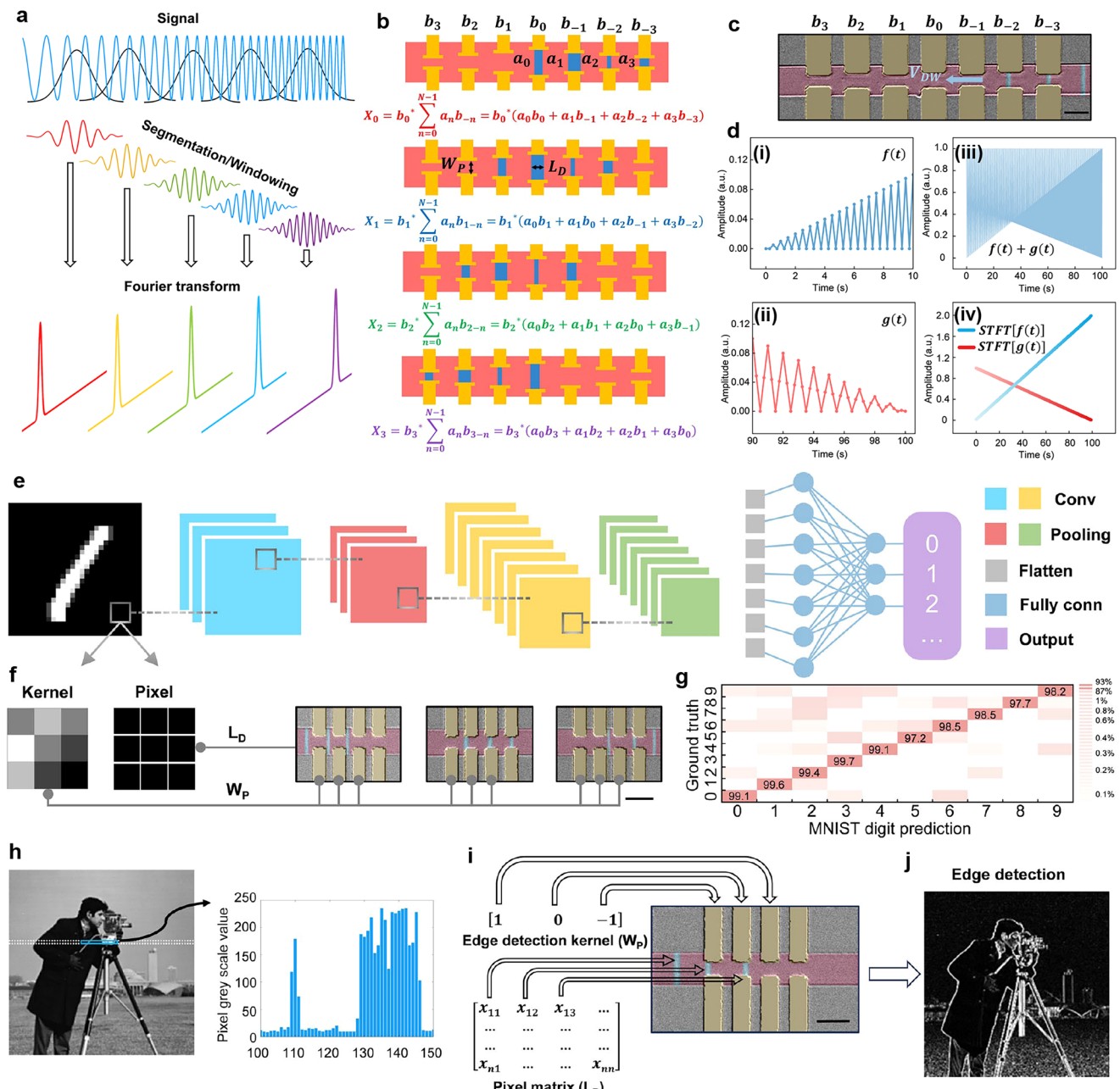

**Fig. 3 | Algorithm implementation on the MCA. a** Conceptual illustration of the STFT. The input signal is segmented and Fourier-transformed. **b** Schematic of 4-point STFT using a 7-Hall-pad MCA device. Input values $a_n$ map to $L_D$, and kernel coefficients $b_{k-n}$ to $W_P$. Sequential domain shifting and AHE readout yield Fourier components $X_n$. **c** Magnetic-domain snapshot as domains shift across the 7-Hall-pad device. Scale bar: 14 μm. **d** Experimental STFT demonstration. (i) Initial segment of $f(t)$. (ii) Final segment of $g(t)$. (iii) Combined input $f(t) + g(t)$. (iv) MCA-generated STFT output showing the temporal evolution of both components. **e** CNN flow diagram for MNIST handwritten digit recognition. **f** Implementation of a $3 \times 3$ convolutional kernel using three MCA devices, each encoding a $1 \times 3$ kernel via $W_P$ configurations (gray dots). Pixel intensities map to $L_D$. Scale bar: 20 μm. **g** Confusion matrix for the MNIST task, showing 98% average accuracy. **h** Left: $256 \times 256$ greyscale 'Cameraman' image. Right: Intensity profile along a line cut; x-axis denotes pixel position. **i** Experimental setup for edge detection using the kernel [1, 0, –1] encoded via $W_P$, with pixel intensities mapped to $L_D$. Scale bar: 20 μm. **j** Edge-detected image produced using the MCA.

undergoes a DFT. The DFT of a windowed signal is given by: $X_k = e^{-\frac{i\pi k^2}{N}} \sum_{n=0}^{N-1}(x_n \cdot e^{-\frac{i\pi n^2}{N}})(e^{\frac{i\pi(k-n)^2}{N}})$, where $x_n$ is the $n$th input signal sample, $X_k$ is the $k$th Fourier-transformed component, and $N$ is the segment length. By defining auxiliary variables $a_n = x_n \cdot e^{-\frac{i\pi n^2}{N}}$ and $b_n = e^{\frac{i\pi n^2}{N}}$, the DFT expression can be recast into a convolutional form:

$$X_k = b_k^* \sum_{n=0}^{N-1} a_n b_{k-n} \qquad (5)$$

Where $b_k^*$ is the complex conjugate of $b_k$.

This reformulation highlights that the DFT computation can be interpreted as a convolution between $a_n$ and $b_{k-n}$, enabling its direct implementation using the MCA platform. Specifically, input signals $a_n$ are mapped to domain lengths ($L_D$), while the convolution kernel coefficients $b_{k-n}$ are mapped to the electrode spacings ($W_P$).

To experimentally demonstrate this capability, we perform a 4-point segmentation (i.e., $N = 4$) STFT using an MCA device. The device implementation is illustrated in Fig. 3b: input segments consisting of four domains ($L_D = a_0, a_1, a_2, a_3$) are shifted across a kernel of

seven Hall electrodes programmed with spacings corresponding to $(W_P = b_3, b_2, b_1, b_0, b_{-1}, b_{-2}, b_{-3})$. Each shift operation advances the domain sequence by one unit, and the Fourier-transformed outputs $(X_0, X_1, X_2, X_3)$ are obtained by summing the AHE voltages across the corresponding electrode configurations. Additional details regarding the STFT mathematical derivation and device implementation are provided in "Methods" and Supplementary Fig. 8.

Following this 4-point STFT configuration (Fig. 3b), we fabricate a 7-Hall-pad MCA device, shown in Fig. 3c. As test signals, we prepare two synthetic time-varying functions: $f(t) = \frac{t}{200}[\cos(4\pi t) + 1]$, representing a signal with increasing amplitude, and $g(t) = \left(0.5 - \frac{t}{200}\right)[\cos(2\pi t) + 1]$, representing a signal with decreasing amplitude. Both signals are sampled at intervals of 0.25 s.

Figures 3di and 3dii display zoomed-in segments of $f(t)$ and $g(t)$, respectively, highlighting their amplitude variations over time. Figure 3diii shows the superposition of $f(t) + g(t)$, which is segmented and input into the MCA device. A snapshot of the domain segment shifting across the 7-Hall-pad structure during operation is captured in Fig. 3c.

After completing the sequential shifting and Hall voltage readout processes, the recorded AHE signals are post-processed (scaled in computer) to construct the STFT result, as shown in Fig. 3div. This result clearly resolves the temporal evolution of the individual frequency components associated with $f(t)$ and $g(t)$.

Finally, as discussed earlier, the fabricated electrode spacings $W_P$ are mapped to the convolution kernel. Variations in $W_P$ can therefore influence the accuracy of the STFT output. This effect is systematically investigated and discussed in Supplementary Note 7.

Additionally, to demonstrate scalability, we implement an 8-point STFT using a 15-Hall-pad MCA device, as presented in Supplementary Fig. 9. By expanding the number of Hall electrodes, the MCA architecture supports longer segmentation windows and higher-resolution STFT calculations, showcasing the potential for scaling the platform toward more complex signal processing tasks.

CNNs are foundational in computer vision, enabling applications such as object detection, image segmentation, and facial recognition. Their versatility extends across diverse fields, including natural language processing, medical imaging, and strategic decision-making tasks. To improve computational efficiency, CNNs frequently rely on specialized hardware accelerators.

The MCA platform is designed to support this role by accelerating convolutional operations with fixed weights. Unlike hardware platforms aimed at training, our device specifically targets inference tasks where pre-trained convolutional layers remain unchanged. This design aligns naturally with the widespread use of transfer learning in contemporary AI workflows.

In standard AI pipelines, transfer learning involves initializing models with pre-trained weights derived from large, diverse datasets. These pre-trained networks extract generalized features applicable across many domains. Fine-tuning is then performed on smaller, task-specific datasets, typically by freezing the early convolutional layers responsible for basic feature extraction (e.g., edges, textures) and retraining only the later layers to specialize the model. This hierarchical training strategy significantly reduces computational overhead, accelerates deployment, and minimizes the size of required datasets.

For example, in large language models like GPT-derived architectures used for code assistance, fine-tuning typically modifies only a small portion of the network—adding a task-specific head—while the majority of the network remains unchanged. In computer vision, a typical application involves adding a recognition head onto a pre-trained backbone such as MobileNetV2. As documented by Google[27], this approach reduces the number of trainable parameters from 2,257,984 to just 1281, cutting training time and energy by over an order of magnitude.

Transfer learning has been shown to consistently outperform training from scratch when working with limited data[28,29], and underpins many state-of-the-art models, including Llama[30] and BERT[31]. It is now the dominant paradigm across AI applications.

Our MCA device is ideally suited for accelerating the fixed convolutional layers that dominate deep neural networks in terms of area, energy consumption, and computation time—often accounting for over 90% of network resources[27]. By offloading these computationally intensive, fixed-weight layers to the MCA platform, overall system performance can be significantly enhanced, reducing the computational footprint, latency, and energy demand compared to traditional CMOS implementations.

For the adaptable, trainable layers toward the end of the network, conventional hardware such as GPUs or application-specific integrated circuits can be employed. This creates a hybrid system architecture that leverages the strengths of both spintronic and CMOS technologies—speed and efficiency from MCA, and flexibility from conventional digital hardware.

To demonstrate the MCA's effectiveness in accelerating CNNs, we apply the platform to a practical task: recognizing MNIST handwritten digits. The system flow diagram is shown in Fig. 3e, where input images undergo a typical CNN pipeline including convolutional, max pooling, flattening, fully connected, and output layers. Additional algorithmic details are provided in "Methods" and Supplementary Fig. 10a.

In our experimental setup, the pre-trained convolutional layers are physically implemented using the MCA device, while the subsequent layers (pooling, fully connected, classification) are executed in silico. As illustrated in Fig. 3f, the convolutional layer is realized by mapping pixel intensity values to domain lengths ($L_D$) and kernel weights to Hall electrode spacings ($W_P$).

Due to the one-dimensional nature of domain shifting, a $3 \times 3$ convolutional kernel must be partitioned into three parallel $1 \times 3$ kernels. These are assigned to three separate MCA devices to perform two-dimensional scanning. In Fig. 3f, three MCA devices are shown, each with three pairs of Hall electrodes (marked by gray dots) forming the $1 \times 3$ kernel structure. As domains representing pixel values are shifted across the devices, AHE voltages are sequentially read and summed to complete the convolution operation. Additional filters can be appended by cascading MCA devices.

Recognition tests were conducted on 10,000 MNIST handwritten digit samples. An average classification accuracy of 98% was achieved, as summarized in the confusion matrix shown in Fig. 3g. The plot is presented on a logarithmic scale to highlight misclassification rates below 1%. Detailed accuracy results are further provided in Supplementary Fig. 10b.

To further assess the practicality and generalization of the MCA platform, a more advanced image-classification task using the CIFAR-10 dataset was also implemented (Supplementary Note 8). In addition, variations in electrode spacing ($W_P$) and domain length ($L_D$)—which introduce slight deviations from ideal linearity—could affect convolution accuracy and, consequently, the overall algorithmic performance. These effects are systematically examined and discussed in Supplementary Note 9.

Convolution is a fundamental operation in image processing, forming the basis for tasks such as edge detection, sharpening, blurring, and averaging. Leveraging the MCA platform's convolutional capabilities, we demonstrate these key image processing functions experimentally.

For the input, we select the standard $256 \times 256$-pixel, 8-bit greyscale 'Cameraman' image, shown in the left panel of Fig. 3h. This image is represented as a greyscale pixel matrix, where each pixel has an intensity value between 0 and 255. As an illustration, the right panel of Fig. 3h displays the greyscale intensity profile along a single line cut through the image.

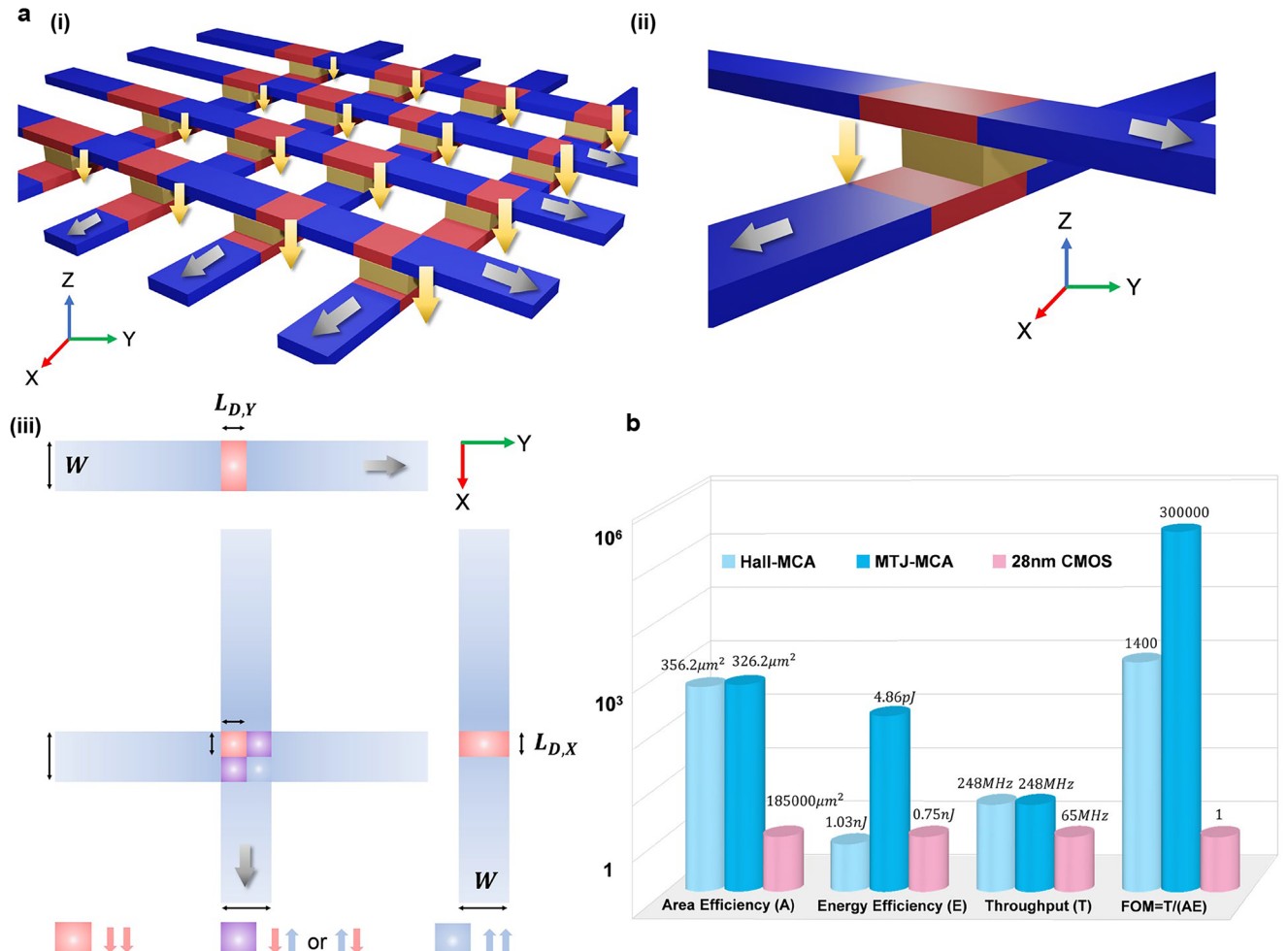

**Fig. 4 | Two-dimensional MTJ-based MCA architecture and benchmarking against CMOS. a** Schematic of the 2D MTJ-based MCA (MTJ-MCA). (i) Conceptual design using crossing arrays of MTJs. Gray arrows indicate domain-shifting currents; yellow arrows denote vertical readout through the MTJs. (ii) Schematic of a single MTJ junction. (iii) Top view of an MTJ junction and overlapping magnetic strips. Red and blue regions represent −**z** and +**z** magnetizations, with domain lengths $L_{D,Y}$ and $L_{D,X}$ along the **y**- and **x**-directed strips and junction width $W$. Overlap regions highlight different domain combinations. **b** Benchmarking comparison between the 1D Hall-based MCA, the 2D MTJ-MCA, and CMOS (28 nm). The plot uses a logarithmic scale. The figure of merit is defined as FOM = $T/(AE)$.

Analogous to the CNN demonstration, each pixel value is mapped to a corresponding domain length ($L_D$) within the MCA device. To perform different convolutional operations, distinct processing kernels are encoded by setting specific spacings between pairs of Hall electrodes ($W_P$).

Figure 3i captures a snapshot during the edge detection experiment. Here, the greyscale values of three adjacent pixels are mapped to three magnetic domains with corresponding $L_D$ values. These domains are shifted sequentially across a kernel designed for edge detection, consisting of three Hall electrode pairs representing the coefficients [1, 0, −1]. In this setup, positive coefficients are connected with standard polarity, negative coefficients are connected with reversed polarity, and zero coefficients are left unconnected.

The resulting edge-detected image, generated using the MCA platform, is shown in Fig. 3j. Additional image processing functions, including sharpening, Gaussian blurring, and averaging, are implemented by programming different kernel configurations into the device. The corresponding results are presented in Supplementary Fig. 11.

### Toward two-dimensional architectures and benchmarking against CMOS

As discussed previously, our one-dimensional (1D) Hall-based MCA platform supports the majority of convolution operations required for pre-trained AI networks and signal processing tasks. However, in scenarios where dynamic coefficient updates are necessary—such as during AI training—more flexible architectures are desirable.

To address this limitation, we propose a two-dimensional (2D) extension of the MCA platform by integrating crossing arrays of magnetic tunnel junctions (MTJs), resulting in a 2D MTJ-MCA architecture (Fig. 4ai, ii). In this design, convolutional inputs and weights are mapped to domain lengths along the bottom and top magnetic strips, respectively. The procedures for weight programming and independent control of the upper and lower magnetic domains are described in Supplementary Note 10 and Note 11, respectively. As illustrated schematically in Fig. 4aiii, the **x**- and **y**-direction magnetic strips contain domains with lengths $L_{D,X}$ and $L_{D,Y}$, while $W$ denotes the lateral dimension of the junction overlap region. Additional details and analysis of the domain configuration of Fig. 4aiii are discussed in Supplementary Note 12.

The readout function for a device with $N$ junctions is formulated by considering an **x**-directed bottom strip overlapped by $N$ **y**-directed top strips. When a bias voltage $V$ is applied across the MTJs, the total current through the bottom strip is given by:

$$I[k] = c_3 VN + c_4 V \sum_{n=1}^{N} \left( \frac{1}{2} W - L_{D,Y}[n] \right) \left( \frac{1}{2} W - L_{D,X}[n-k] \right) \quad (6)$$

where $c_3$ and $c_4$ are device-dependent constants, and $k$ denotes the clock cycle corresponding to domain shifting. As shown in Eq. (6), the output current naturally results from a convolution operation, with both the input and kernel coefficients being tunable via domain length modulation. A detailed derivation of this readout mechanism is provided in "Methods".

We next present a comprehensive benchmarking analysis comparing the MCA platform against standard CMOS technology, using experimental parameters or experimentally derived estimates. The evaluation focuses on three key metrics—area, throughput, and energy consumption—for an 8-bit, 1024-point convolution task, targeting a 28 nm process node.

The results are summarized in Fig. 4b, with detailed calculations provided in the Supplementary Note 13.

The sampling and summation circuits for the MCA were synthesized using a commercial 28 nm process design kit (PDK), and their area and energy consumption were extracted accordingly. Notably, in the MCA architecture, the overall area is dominated by the sampling circuitry (comprising five transistors per cell), accounting for ~70% of the total footprint. For the CMOS benchmark, a full synthesis application-specific integrated circuit (ASIC) targeting an equivalent convolutional operation was performed using the same PDK. Details of the CMOS circuit design and benchmarking methodology are provided in the Supplementary Note 14.

Benchmarking results (Fig. 4b) reveal that the MCA platform offers substantial improvements in both area and throughput compared to CMOS. This advantage arises from the highly compact physical implementation of the shifting and multiply–accumulate (MAC) operations, which would otherwise require hundreds of transistors per unit in CMOS.

On the other hand, as shown in Fig. 4b, the current energy consumption per operation in the 1D Hall-based MCA is slightly higher than that of CMOS, primarily due to the large current densities required to drive high-speed domain wall motion. However, previous studies[22] indicate that optimizing material growth and minimizing pinning sites can significantly reduce the required shift current, thereby enhancing the energy efficiency of future 2D MTJ-based designs. Compared with optical systems for physical-convolution computing[32,33], the MCA exhibits a comparable level of energy consumption but higher latency, primarily due to the lower speed of domain-wall motion. Nonetheless, its area footprint is orders of magnitude smaller, offering clear advantages for large-scale integration and on-chip deployment. The crosstalk behavior of the MCA in large-scale array configurations is also investigated to assess integration feasibility, as discussed in Supplementary Note 15.

Considering the figure of merit (FOM), defined as:

$$\text{FOM} \equiv \frac{\text{Throughput}\,(T)}{\text{Area}\,(A) \times \text{Energy}\,(E)} \left(\text{FOM} = \frac{T}{AE}\right) \qquad (7)$$

The MCA platform demonstrates a 3–5 orders of magnitude improvement compared to CMOS technology at the same process node, as highlighted in Fig. 4b.

## Discussion

We demonstrate a hardware platform capable of performing convolution through sequential magnetic domain-wall motion, with applications spanning deep learning, computer vision, and signal and image processing. Benchmarking shows performance improvements of three to five orders of magnitude over 28 nm CMOS technology. By directly mapping magnetic-domain dynamics onto convolutional operations, the MCA unifies data storage and computation within a nonvolatile physical framework, establishing a spintronic compute-in-memory architecture. The experimental realization of convolution—a core operation in modern machine learning—thus represents a key step toward general-purpose spintronic accelerators beyond Boolean logic.

While the current prototype demonstrates the viability of MCA-based convolution, several factors motivate future development. Pinning variations contribute to elevated energy consumption and occasional stochasticity in DW motion, highlighting the importance of improved material uniformity and controlled pinning landscapes. In addition, the present one-dimensional, fixed-weight structure limits kernel programmability; transitioning to a two-dimensional MTJ-MCA array with electrically tunable weights will be necessary for more versatile compute-in-memory functions.

Looking forward, opportunities for enhancing MCA performance arise at both the architectural and materials levels. Architecturally, expanding to two-dimensional arrays will support larger datasets and more complex workloads[34], and the MCA's simple DW-based racetrack geometry[35] facilitates integration with CMOS back-end processes. On the materials side, strong-DMI multilayers[21] and ferrimagnets[24] offer pathways to sub-10 nm domain scaling[21], high on–off ratios, and ultrafast operation[24], while optimized pinning landscapes may further reduce DW driving energy[22]. Realizing these benefits will require careful integration, including maintaining stable alternating-domain configurations and ensuring compatibility with MTJ readout stacks for high-TMR sensing.

Overall, this work positions the MCA as a promising foundation for scalable, energy-efficient spintronic computing, with clear opportunities for continued advancement through innovations in materials, device engineering, and system integration.

## Methods

### Thin film deposition and device fabrication

Thin films composed of Ta (5 nm)/$Co_{40}Fe_{40}B_{20}$ (1 nm)/MgO (2 nm)/Ta (2 nm) were deposited onto semi-insulating silicon substrates covered with a 100 nm thermal oxide layer. Deposition was performed at room temperature using both direct current (DC) and radio-frequency (RF) magnetron sputtering in an AJA sputtering system. The base pressure during deposition was maintained below $1 \times 10^{-8}$ torr, with an argon working pressure of 3 mtorr. Post-deposition, the films underwent vacuum annealing at 250 °C for 30 min to enhance perpendicular magnetic anisotropy (PMA), as confirmed in Supplementary Fig. 2. Exposure to ambient conditions naturally oxidized the top 2 nm Ta layer into TaOx, providing environmental protection. Device patterning was achieved through a combination of photolithography, dry etching, wet etching, and metal evaporation processes. While the overall fabrication flow resembled that of a conventional Hall bar device, modifications were introduced to integrate an Oersted-field generating microwave strip for domain nucleation and a Hall pad–Via structure for precise control of the Hall pad separation distance ($W_P$). The detailed fabrication sequence is illustrated in Supplementary Fig. 3.

### MOKE microscopy characterization

Wide-field Magneto-Optic Kerr Effect (MOKE) imaging was performed using a custom-designed MOKE microscope setup, providing spatial resolution of ~360 nm and temporal resolution of 20 ms. An external magnetic field was generated by a GMW-5201 Helmholtz coil system, powered through a Kepco BOP 5-20D power supply. Initially, the sample was saturated under a strong negative out-of-plane magnetic field ($-H_z$), and a reference background image was recorded. Subsequent magnetic images were obtained by subtracting this background, enabling clear magnetic domain contrast. To prepare the initial domain configurations used in Fig. 2d, e, the MCA devices were first saturated under a large $-H_z$. After removing the field, the magnetic layer retained a uniform negative out-of-plane magnetization ($-m_z$). MOKE imaging sequences were then captured during the processes of domain nucleation and subsequent domain wall motion.

**Domain nucleation, domain motion, and AHE voltage detection**

After initializing the MCA device into a uniform $-m_z$ magnetization state, domain nucleation was triggered by sending a current pulse through the Oersted field line, sourced from a Keithley 2636A SourceMeter. The generated $+m_z$ domain length ($L_D$) was controllably adjusted by tuning the pulse parameters, including amplitude, polarity, and duration. Following nucleation, domains were displaced to designated locations to reserve space for the creation of additional domains. This domain shifting process was driven by applying a series of current pulses (1 mA) to the Shifting channel using an Agilent 33250A arbitrary waveform generator. Successive domains were similarly created and spaced to align precisely with the intended Hall sensing regions. After forming the desired domain array, all domains were shifted simultaneously into the Hall detection region. AHE measurements were carried out sequentially, corresponding to the convolution operation order, as demonstrated in Fig. 2e. Specifically, the transverse Hall voltage ($V_{AHE}$) was measured whenever a domain resided within the Hall sensing channel. The typical measurement configuration is presented in Supplementary Fig. 2b, with a constant current $I$ (0.1 mA) applied along the device and $V_{AHE}$ monitored across the transverse electrodes.

All experiments were conducted at ambient room temperature.

**Analytical derivation of Hall-MCA convolution operation**

The MCA device is structured such that the AHE voltage ($V_H$) can be expressed as:

$$V_H = \alpha' \cdot M_z \cdot I_H \cdot \frac{W_P}{W} \qquad (8)$$

Here, $\alpha'$ represents a material-dependent constant, $M_z$ is the net out-of-plane magnetization within the sensing region between a pair of Hall electrodes, $I_H$ is the applied current along the +**x** direction, $W_P$ is the separation between the two Hall electrodes, and $W$ is the total device width.

The net magnetization $M_z$ is described by:

$$M_z = \left( -2\frac{L_D}{L_P} + 1 \right) M_s \qquad (9)$$

where $M_s$ denotes the saturation magnetization, $L_D$ is the length of the reversed ($-z$) magnetic domain, and $L_P$ corresponds to the contact length of the Hall electrodes.

Initially, the Hall sensing area is uniformly magnetized along the +**z** direction. When a reversed domain of length $L_D$ enters the region, the net $M_z$ decreases linearly with $L_D$. As $\alpha'$, $M_s$, and $W$ are intrinsic material and device constants, they can be consolidated into a single coefficient $\alpha$ for simplicity.

Substituting the expression for $M_z$ into the $V_H$ formula yields:

$$V_H = \alpha I_H (2L_D - L_P) W_P \qquad (10)$$

This result corresponds to Eq. (1) presented in the main text.

**Fundamental operating principles of the MCA**

The realization of the MCA relies on four fundamental components: (1) Establishment of PMA, ensuring the preferential alignment of magnetization along the z-axis; (2) Fine control over domain length, critical for accurate input encoding; (3) DW displacement driven by SOT, necessary for sequential domain shifting; and (4) Electrical readout of the convolution signal via AHE voltage measurements.

All these elements are implemented in a well-established material system comprising a Heavy Metal/Ferromagnet/Insulator heterostructure—specifically, Ta/CoFeB/MgO—as depicted in Supplementary Fig. 1 and Supplementary Fig. 2.

For (1), PMA is achieved through interfacial effects between the CoFeB and MgO layers[36], as verified experimentally (Supplementary Fig. 2).

For (2), domain and DW formation is accomplished through Oersted fields generated by a transverse conducting stripline (Supplementary Fig. 1a). In this schematic, blue and red regions denote $+m_z$ and $-m_z$ magnetization, respectively, while the white area represents the DW transition zone. By injecting a short current pulse along the +**y**-axis into the strip, a localized Oersted ($B$) field (green arrows) is produced, nucleating a $-m_z$ domain. The domain length ($L_D$) can be tuned by adjusting the current pulse parameters—with a positive (negative) current increasing (decreasing) $L_D$.

For (3), DW motion is induced by SOTs generated through spin Hall effects[9,26] within the Ta underlayer. When a current $j$ flows along the +**x**-direction (Supplementary Fig. 1b), spin accumulations ($\sigma_{+y}$ and $\sigma_{-y}$) build up at the top and bottom surfaces of the Ta layer, as shown in the right-hand schematic. These spins act as effective **z**-directed fields on the DW moments $m$ and exert a torque described by:

$$H_{SOT}^Z \sim m \times \sigma_{+y} \qquad (11)$$

Due to the right-handed DMI[15,16,37] at the Ta/CoFeB interface[9,15], the DWs possess chiral structures that respond differently to $\sigma_{+y}$. The effective fields $H_{SOT}^Z$ act oppositely on the two DWs (since the angle between $m$ and $\sigma_{+y}$ is $\pm 90°$), satisfying:

$$H_{SOT}^Z \sim \sin(\pm 90°) \sim \pm z \qquad (12)$$

As a result, SOTs drive both DWs coherently along the +**x**-direction with velocity $V_{DW}$[9,15]—a mechanism commonly referred to as SOT-driven DW motion.

For (4), the AHE readout leverages the intrinsic properties of ferromagnetic metals such as CoFe and CoFeB. In the Ta/CoFeB/MgO structure, the AHE resistance $R_{AHE}$ is typically around $1 \Omega$[9], as demonstrated in Supplementary Fig. 2. This AHE voltage provides a direct and efficient means of electrically sensing the domain configurations necessary for convolution operations.

**Simulation and optimization of the Oersted field channel using COMSOL**

To optimize the performance of the gold striplines (Oersted field channels) and reduce energy consumption, a COMSOL Multiphysics simulation study was conducted. The target was to generate an out-of-plane magnetic field of 50 Oe—matching the coercivity of the material—at a distance of 75 nm from the sample boundary, in accordance with device constraints. The study involved tuning three main control parameters: stripline thickness, width, and applied current. By carefully adjusting these variables, we aimed to achieve the required magnetic field strength while minimizing power dissipation. The optimization targeted two wire lengths, $l = 346$ nm and $l = 75$ nm, which reflect practical design constraints (Supplementary Fig. 4a). The optimization process employed the Nelder-Mead algorithm within the Magnetic and Electric Fields module of COMSOL. The objective function was set to minimize the volumetric loss density (expressed in watts). Initial conditions for the optimization were chosen as: wire width $w = 1$ nm, thickness $t = 1$ nm, and starting current $i_0 = 0.2$ mA. Control parameter boundaries were defined between 1 nm and 300 nm for both width and thickness, and between 0.05 mA and 5 mA for current. Upon convergence, the optimal parameters for the 346 nm-long wire were determined to be a thickness of 156 nm, width of 104 nm, and applied current of 3.8 mA, resulting in a total power consumption of less than 7 μW. For the 75 nm-long wire, the optimal configuration was a thickness of 38.5 nm, width of 108 nm, and current of 2.9 mA, with energy dissipation reduced to below 4 μW. The magnetic field spatial distribution and the optimization progression

(power loss versus iteration number) for the 346 nm wire are presented in Supplementary Fig. 4b, while corresponding results for the 75 nm case are shown in Supplementary Fig. 4c.

### Device design and dimensioning for MCA functions

The MCA device architecture was engineered to support different convolution operations − including Gaussian blurring, image averaging, sharpening, and STFT−as detailed in Supplementary Fig. 6. The starting point of the design involved defining the spacing between adjacent Hall electrodes ($W_P$) according to the intended convolution kernel. For example, in the Gaussian blurring function, the sequence of $W_P$ values was set to 3–12–18–12–3 µm, providing an approximation of a Gaussian curve. The Hall electrode contact length ($L_P$) was selected to be either 14 µm or 20 µm, balancing the need for fine domain length control with fabrication feasibility. A Hall pad width ($W_H$) of 6 µm was chosen to maximize successful Via fabrication rates. To maintain reliable photolithographic patterning, Hall pads were positioned 3 µm inward from the magnetic strip edge. Based on these considerations, the overall magnetic strip width ($W_M$) was determined as $W_M = 18 + 6 × 2 + 3 × 2 = 36$ µm. For domain nucleation, the Oersted field channel width ($W_O$) was designed at 15 µm. This allows for the formation of a magnetic domain under a 50 Oe coercive field when applying a 200 mA pulse. Further tuning of $W_O$ can be performed if higher field magnitudes are required. Similar design strategies were applied to devices optimized for averaging, sharpening, and STFT functionalities. Dimension details for these kernels are summarized in Supplementary Fig. 6.

### STFT formulation and realization with MCA devices

The DFT is a mathematical operation that converts a sequence of time-domain samples into a corresponding set of complex-valued frequency-domain coefficients. It can be expressed in a convolution-like form, enabling its hardware implementation via sequential domain shifting in the MCA device.

The DFT is formally defined as:

$$X_k = \sum_{n=0}^{N-1} x_n \cdot e^{-\frac{i2\pi}{N}kn} \tag{13}$$

Where $x_n$ are the input time-domain samples and $X_k$ are the resulting frequency-domain outputs.

This equation can be algebraically rearranged into a form that resembles a convolution:

$$X_k = e^{-\frac{i\pi k^2}{N}} \sum_{n=0}^{N-1} \left( x_n \cdot e^{-\frac{i\pi n^2}{N}} \right) \left( e^{\frac{i\pi(k-n)^2}{N}} \right) \tag{14}$$

By introducing the substitutions $a_n = x_n \cdot e^{-\frac{i\pi n^2}{N}}$ and $b_n = e^{\frac{i\pi n^2}{N}}$, the expression becomes:

$$X_k = b_k^* \sum_{n=0}^{N-1} a_n b_{k-n} \tag{15}$$

This transformation highlights a convolution structure involving a fixed kernel $b_{k-n}$. Importantly, "fixed kernel" implies that the coefficients $b_{k-n}$ are independent of the input signal $x_n$ and remain unchanged. In the MCA device, this feature is leveraged by encoding these fixed weights through predefined electrode spacings ($W_P$) during fabrication.

Since DFT outputs are complex-valued, the convolution must be separated into real and imaginary components. The input signal, pre-multiplied by the complex phase factor $e^{-\frac{i\pi n^2}{N}}$, is first decomposed into real and imaginary parts. Each part is then individually mapped to the domain lengths ($L_D$) in separate MCA devices. The real and imaginary parts of the convolution kernel $b_{k-n}$ are similarly mapped to dedicated

devices. Convolution operations proceed independently within these MCA devices by sequential domain shifting and AHE readout. After completing the convolution across four MCA devices (two for the input real/imaginary parts and two for the kernel real/imaginary parts), the results are recombined by post-multiplying by $b_k^*$. The final calculated values $X_k$ reveal the amplitude and phase information across different frequency components, as depicted in Supplementary Fig. 8.

### Implementation of CNN inference using MCA devices

To assess the performance of the MCA-based CNN under varying input and weight precision constraints, we conducted the following experimental procedure: The original MNIST handwritten digit dataset, consisting of 28 × 28-pixel 8-bit grayscale images, was first quantized according to the specified input precision (e.g., n-bit resolution). Each pixel value was rescaled to the nearest quantization level between 0 and 1 corresponding to the chosen bit depth. A CNN architecture comprising two convolutional layers, two fully connected layers, and two max-pooling layers (shown schematically in Fig. 3e) was trained on the training set using the ADAM optimizer for 10 epochs. Full-precision (floating-point) weights were used during initial training to establish baseline performance on the test set. For network quantization, each layer's weights were mapped into discrete steps determined by the weight precision constraint: the 1st and 99th percentile values of the trained weights were identified, and the range between them was divided evenly according to the number of quantization steps. Each original weight was then projected onto its closest discrete value, and the resulting quantized model was evaluated on the test set to determine the ideal n-bit weight accuracy. To simulate hardware-based inference, we constructed a model where each convolutional operation was replaced by experimental data collected from the MCA devices. The filter (kernel) coefficients were taken from the experimental measurements shown in Fig. 2f. The inputs were mapped using the linearized AHE voltage responses ($V_{AHE}$) obtained in Supplementary Fig. 7 to account for real device behavior. The full evaluation workflow for this MCA-based CNN implementation is summarized in the flowchart in Supplementary Fig. 10a. The final recognition accuracies, averaged across multiple trials, were plotted as functions of input and weight precision, with the results shown in Supplementary Fig. 10b.

### Derivation of convolution operation for MTJ-based MCA devices

In a single MTJ unit, the parallel (P) and antiparallel (AP) alignments of the magnetic layers yield tunneling resistances of R and 2R, respectively, due to the tunneling magnetoresistance (TMR) effect. When two magnetic domains, with respective lengths $L_{D,X}$ and $L_{D,Y}$, overlap at the junction area (of total width $W$) as depicted in Fig. 4a(iii), the structure can be modeled as four parallel-connected sub-junctions with effective areas:

- $A_1 = L_{D,X} × L_{D,Y}$ (P) (P-aligned region)
- $A_2 = (W - L_{D,X}) × L_{D,Y}$ (AP-aligned region)
- $A_3 = (W - L_{D,Y}) × L_{D,X}$ (AP-aligned region)
- $A_4 = (W - L_{D,X}) × (W - L_{D,Y})$ (P-aligned region)

The overall conductance (G) of the junction can thus be expressed as the sum of the conductances of these four regions:

$$G = G_1 + G_2 + G_3 + G_4 = \frac{1}{R\frac{W^2}{A_1}} + \frac{1}{2R\frac{W^2}{A_2}} + \frac{1}{2R\frac{W^2}{A_3}} + \frac{1}{R\frac{W^2}{A_4}} \tag{16}$$

Substituting the expressions for the areas and simplifying yields:

$$G = \frac{1}{RW^2}\left[ \left(\frac{1}{2}W - L_{D,X}\right)\left(\frac{1}{2}W - L_{D,Y}\right) + \frac{3}{4}W^2 \right] \tag{17}$$

This can be further written as:

$$G = c_3 + c_4 \left( \frac{1}{2} W - L_{D,X} \right) \left( \frac{1}{2} W - L_{D,Y} \right) \quad (18)$$

where $c_3$ and $c_4$ are constants that depend on the device parameters.

To extend this model to a full device containing $N$ MTJ junctions, consider a structure where a horizontal (**x**-direction) bottom magnetic strip overlaps with $N$ vertical (**y**-direction) top strips. Applying a bias voltage $V$ across the device results in a total current $I[k]$ along the bottom strip, which is the cumulative contribution from each junction. The current readout for such an $N$-junction device is formulated as:

$$I[k] = c_3 V N + c_4 V \sum_{n=1}^{N} \left( \frac{1}{2} W - L_{D,Y}[n] \right) \left( \frac{1}{2} W - L_{D,X}[n-k] \right) \quad (19)$$

This corresponds to Eq. (6) presented in the Main Text.

## Data availability

All data needed to evaluate the conclusions in the paper are available within the article and its Supplementary Information files. All data generated during the current study are available from the corresponding author upon request.

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

## Acknowledgements

B.D., T.W., A.L., S.X., C.C., K.W., D.L., M.J., Y.C., P.H., Y.L., Q.S., H.R.H., L.T., H.S.H., and K.L.W. acknowledge the support from the National Science Foundation (NSF) Award No. 1810163 and No. 1611570; and the Army Research Office Multidisciplinary University Research Initiative (MURI) under grant numbers W911NF16-1-0472 and W911NF-19-S-0008.

C.C. and T.C. acknowledge the support from the National Science and Technology Council (NSTC), Taiwan Ministry of Science and Technology (MOST) Grant No. 107-2628-E-009-003-MY3 and 110-2221-E-A49-081-MY2. C.Y. and Y.H. acknowledge the support from the National Key Research and Development Program of China under Grant No. 2022YFA1203902 and 2022YFA1204003, and National Natural Science Foundation of China under Grant No. 52473263 and 52201287.

## Author contributions

B.D., T.W., and A.L. designed, planned, and initiated studies. B.D., T.W., and S.X. prepared material samples. B.D., T.W., and D.L. conducted the MOKE and transport measurements and analyzed the data. K.W. and C.C. fabricated the devices. M.J. performed the COMSOL simulation. Y.H. and K.L.W. supervised the project. B.D., T.W., A.L., Y.H., and K.L.W. drafted the manuscript. All authors discussed the results and Y.C., P.H., Y.L., C.Y., Q.S., H.R.H., L.T., H.S.H., and T.C. commented on the manuscript.

## Competing interests

The authors declare no competing interests.
