## [Transparent Peer Review file · Nature Communications]

Domain Wall Motion-Driven Magnetic Convolutional Accelerator

Corresponding Author: Professor Kang Wang

Version 0:

Reviewer comments:

Reviewer #1

(Remarks to the Author)

In this paper, Dai et al designed a magnetic convolutional accelerator (MCA) device based on current-driven domain-wall motion. They fabricated several μm -scale MCA devices and demonstrated the basic performance of domain-wall sensing and convolution computing. During the operation, the information is encoded in domain patterns and read via Hall voltages with various geometric parameters. The manuscript is generally well written and the concept of convolutional computing using DW motion is interesting. However, I have several concerns regarding the feasibility of such domain-wall MCA devices, which must be addressed before the manuscript can be considered for Nature Communications.:

1. The performance of current-induced domain-wall displacement seems too idealistic. The authors encode the input signal in the domain length. But the domains are prone to getting deformed due to the presence of pinning sites. Especially, the Hall channel may act as the geometric pinning sites. Since input variations can affect the functionalities, the domain-wall pinning may significantly degrade the performance of MCA devices. The authors should present the stability test of domain size when transferring through the racetrack channel and estimate the influence of domain-wall pinning in downscaled devices. They should also provide a MOKE video to show the smooth domain-wall movement in MCA devices.
2. Though the velocity of current-driven domain-wall motion can be fast, the detection of Hall voltage is through DC measurement in this work, which slows down the operation speed. The author should demonstrate a fast way to read the Hall signal to match the fast domain-wall velocity. In addition, how is the domain-wall velocity in this work? They should measure the domain-wall velocity as a function of current density.

Besides, some other issues need to be addressed:

1. The device used by the authors is based on the ferromagnet CoFeB. The velocity of ferromagnetic DW motion is usually limited to a few hundred meters per second. The authors should provide more experimental evidence to support the expected operating speed of THz. Moreover, the size of the domain wall in ferromagnets is usually tens of nanometers. The authors should provide more evidence to support the expected domain size of 1 nm.
2. They applied a high current to displace the domain wall, which may induce significant Joule heating, particularly when operating at a high frequency. The authors should consider this in their modelling and estimate the device temperature due to Joule heating.
3. There have been several reports using physical systems for convolution computing. The author should compare with other systems and clarify the advantages of MCA devices.
4. The convolutional algorithm relies on the fact that the Hall voltage is linear with respect to the length of domains, whereas there are nonlinear deviations in the real devices (Fig. 2f and 2h). The author should analyze how this affects the algorithm.
5. The authors designed several MCA devices for different kernel functionalities. What is the principle for designing the MCA devices? How does the influence of the Hall probe number?
6. It is not clear to me how they perform the handwritten digit recognition and image edge detection tasks. Which results were performed in physical devices and which results were performed in computers? They should provide more details, such as how to encode the grey scale value into domain length, how to choose the current density and how to correct the offset Hall voltage.

Reviewer #2

(Remarks to the Author)

Authors proposed a nice idea having the weights mapped as spacing between electrode pairs during fabrication, which mean weights are fixed at that stage. The inputs, however are applied as domain lengths that can be dynamically controlled and changed at runtime. However, only this part of the device has been experimentally validated. At this stage, the work has multiple practical issues, which are not resolved or discussed. Furthermore, there are theoretical ideas which are not validated in depth, such as, 2D version where both the weights and inputs can be dynamically programmed.

The critical points are highlighted below.

The manuscript acknowledges that the current 1D Hall-based MCA requires relatively high current densities to drive domain wall motion, leading to energy per operation that is slightly higher than CMOS. While the authors suggest this can be mitigated by material optimization, this is left as future work.

The device relies on precise domain nucleation, length control, and synchronized wall motion. The paper shows impressive lab-scale control but does not address large-array scaling. Scalability and uniformity are critical for any practical accelerator.

While a 2D MTJ-MCA extension is proposed, it is theoretical. Can the authors clarify whether any test structures for the 2D MTJ-MCA have been fabricated?

AHE voltage readout relies on linear mapping of domain length and electrode spacing, but fabrication imperfections introduce nonlinearity. Readout fidelity is fundamental to inference accuracy. Can the authors provide quantitative measurements of deviation from ideal linearity? How does readout noise impact classification accuracy in their CNN demonstration?

The device requires precise deposition of Ta/CoFeB/MgO stacks with PMA, fine Hall electrode spacing, and integration of Oersted nucleation lines. Scalable production is necessary for any hardware accelerator. What is the measured yield of fabricated devices? How sensitive is performance to lithographic variations?

While small-scale devices are demonstrated, large-scale arrays needed for real AI workloads are not. Demonstrating a few cells is valuable, but deployment requires large arrays. How does domain wall synchronization degrade over distance? What strategies exist to handle crosstalk or variation in large arrays?

Can the authors clarify what CMOS baseline design was used? Have they benchmarked against state-of-the-art low-power MAC implementations?

The main application demonstrated is MNIST digit classification (28×28 images). A simple dataset can overstate practical readiness.

The 2D extension is presented analytically, but no experimental validation is shown. Proposing a 2D solution is important, but feasibility is unclear. What is the plan for fabricating test 2D MTJ cells? How will the authors address alignment, yield, and TMR variation across large junction arrays? What is the estimated energy/area/throughput of such 2D designs versus 1D Hall MCA?

Any practical use will require integration with CMOS for control, ADC/DAC, and post-processing. Have the authors analyzed the latency and energy of these conversions? How does this impact the claimed system-level advantage?

Reliability is critical for deployment. High-current pulses may cause electromigration or heating. Have the authors measured temperature rise during operation? What is the estimated lifetime under continuous cycling? Are there mitigation strategies (e.g., duty cycling, material changes)?

Reviewer #3

(Remarks to the Author)

Dai et al present a spintronic convolutional computing platform that exploits DW motion to perform both computation and data storage, realized through Magnetic Convolutional Accelerator (MCA). They demonstrate a concrete advance toward the vision of a general-purpose computing platform or full CIM realization, enabling applications such as short-time Fourier transforms, convolutional neural networks, and image processing. The demonstrated MCA pioneered in convolutional acceleration while simultaneously integrating memory and computation through spintronic mechanisms. Realization of data processing tasks rely on analogies between domain shifting and sliding-window operations, which offers exceptional scalability, energy efficiency, operation speeds and nonvolatility. I recommend publication if authors can address the following concerns.

1. Is the 1D MCA convolution kernel multiplication and Fourier transform implemented through peripheral circuits? How is it realized?

2. The accumulated errors during multiple domain wall movements were not addressed. Such errors pose a significant challenge in physical hardware implementations of multiply-accumulate (MAC) operations.

3. In a 2D MTJ-based MCA architecture, are the magnetic domain regions in the strips used as convolution kernel

coefficients continuously generated in a certain time sequence, or are they fixed within the MTJ region? If they are generated dynamically, how is the simultaneous appearance of magnetic domains in the upper and lower strips within the MTJ region controlled? If they are fixed, how can the corresponding magnetic domain widths be modified for different cases? Does the read current used to measure the MTJ magnetoresistance have any effect on the magnetization direction of the magnetic domains?

4. The fixed-weight approach is only suitable for inference, not for the computationally intensive and diverse training phase. Additionally, the 2D architecture lacks experimental validation.

5. In line 104 and 105, the authors depict 'I_H is the applied current flowing along the +x axis, L_D is the domain length aligned along the -z direction'. It seems not the case as illustrated in Fig. 1b.

6. In Equation 1, the length of the domain wall is not taken into consideration. Will the DW length influence the final convolutional result?

7. Since the domain motion is driven by SOT of the underlying heavy metal, will the external magnetic field used for AHE signal read out affect the domain motion?

8. In Fig. 2f, the y-intercept of V_{AHE} versus W_P seems not zero (about 0.5 mV). What is the reason and can it be adjusted or eliminated?

9. In Fig. 2h, the legend of the picture is incomplete with the absence of corresponding symbol or color.

10. In Fig. 3c, the distinction of fabricated W_P is not evident. Will it influence the mapping to convolutional kernel?

11. In Fig. 4a_{iii}, the red, blue and purple squares can coexist simultaneously in the crossing area or is just a phenological illustration of all possible cases?

12. As claimed in line 362 and 363, energy consumption per operation in the 1D Hall-based MCA is primarily due to the large current densities required to drive high-speed domain wall motion. Will adoption of magnetic solitons like skyrmions or driving method like spin wave in insulators facilitate reduction of energy consumption?

Version 1:

Reviewer comments:

Reviewer #1

(Remarks to the Author)

Overall, the authors have carefully addressed my major concerns and strengthened the work with new data, making the revised manuscript more rigorous. The work highlights the potential of MCA for convolutional computing, especially in energy and area efficiency, contributing to emerging computing paradigms. While most issues are resolved, I have a few minor questions for further clarification or refinement:

1. While the authors correctly note that advanced material systems (e.g., ferrimagnets, SOC materials) could enhance performance, the challenges of integrating these into a functional MCA platform, such as interface engineering and read/write compatibility, are not clearly addressed. A brief acknowledgement of these integration challenges as a direction for future work would strengthen the discussion and enhance its rigor.

2. The conclusion section primarily highlights the benefits and potential research directions of the MCA, but provides limited discussion on its inherent limitations (e.g., energy consumption, constraints of the 1D design, stochastic DW motion). Could a short discussion be added to the discussion or conclusion addressing the current or potential limitations and the challenges that need to be overcome for using MCA as a compute-in-memory device?

3. Some figure captions (for example, Fig. 1c) appear quite lengthy and largely restate content already covered in the main text. Could these captions be rephrased or more concise?

Reviewer #2

(Remarks to the Author)

Authors have successfully addressed the prior review comments. I am glad with the current version of the manuscript and recommend acceptance.

Reviewer #3

(Remarks to the Author)

The authors have adequately addressed the points raised by the referees. I therefore recommend acceptance of the manuscript in its current form.

Version 2:

Reviewer comments:

Reviewer #1

(Remarks to the Author)

The authors have answered my questions. I am happy to recommend its publication in Nature Communications.

Reviewer #1 (Remarks to the Author):

In this paper, Dai et al designed a magnetic convolutional accelerator (MCA) device based on current-driven domain-wall motion. They fabricated several μm -scale MCA devices and demonstrated the basic performance of domain-wall sensing and convolution computing. During the operation, the information is encoded in domain patterns and read via Hall voltages with various geometric parameters. The manuscript is generally well written and the concept of convolutional computing using DW motion is interesting. However, I have several concerns regarding the feasibility of such domain-wall MCA devices, which must be addressed before the manuscript can be considered for Nature Communications.:

Reply:

We sincerely thank the reviewer for the careful evaluation and constructive feedback, which have greatly improved the rigor and clarity of our manuscript. In response to the reviewer's comments, we have conducted additional experiments, micromagnetic simulations, and algorithmic validations to strengthen the technical foundation and address the key feasibility concerns. These new results further confirm the robustness and practicality of the proposed MCA concept. We have extensively revised the Main Text and Supplementary Materials to integrate these new results and enhance the overall clarity and coherence of the manuscript.

Detailed point-by-point responses are provided below:

1. The performance of current-induced domain-wall displacement seems too idealistic. The authors encode the input signal in the domain length. But the domains are prone to getting deformed due to the presence of pinning sites. Especially, the Hall channel may act as the geometric pinning sites. Since input variations can affect the functionalities, the domain-wall pinning may significantly degrade the performance of MCA devices. The authors should present the stability test of domain size when transferring through the racetrack channel and estimate the influence of domain-wall pinning in downscaled devices. They should also provide a MOKE video to show the smooth domain-wall movement in MCA devices.

Reply:

We thank the reviewer for this insightful comment. Our response is divided into two sections:

1. Stability test of domain size during domain transfer and MOKE video of smooth motion
2. Influence of domain-wall pinning in downscaled devices

Stability test of domain size during domain transfer and MOKE video of smooth motion

(a) Quantification of domain size

To examine domain-size stability during propagation, we used MOKE microscopy to directly quantify the domain length as the domains moved along the racetrack channel. The domain size is defined as $length \times width$; since the width is fixed by the channel geometry, only the domain length can vary. Thus, we focused on analyzing variations in domain length during transfer.

We experimentally prepared two representative cases: a wide domain and a narrow domain. The results are shown in **Fig. R1**. For wide-domain propagation, the measured domain lengths exhibit a standard deviation of **0.15 μm (1.9%)**, and for narrow domains, **0.10 μm (2.1%)**. These results confirm the high stability of domain-wall motion.

Revisions:

This section is added into the Supplementary Fig. S3, highlighted in yellow.

A description is added into the Main Text: From Line 188 to 191, highlighted in yellow.

(b) MOKE video of smooth movement

The MOKE video is provided in the **Supplementary Data file**. It shows smooth domain propagation across the MCA device, with minimal influence from the Hall channels or other pinning sites.

We note that slight shape distortions occur during motion; however, the net domain area (size) remains nearly conserved. As a result, the encoded input value remains stable with only minor variations. These small variations have a negligible impact on the AI inference performance — the system still achieves **98.94% accuracy** in handwritten digit recognition. A detailed analysis of this aspect is provided in **Comment 4**.

Influence of domain-wall pinning in downscaled devices

We performed additional micromagnetic simulations on devices with dimensions of **256 × 64 × 1 nm³** and **1024 × 256 × 1 nm³** (**Fig. R2ai**). Following established methods [1,2], pinning was modeled by introducing grains, as shown in **Fig. R2aii**. The grain size and associated material parameters—including perpendicular magnetic anisotropy, saturation magnetization, and

Dzyaloshinskii–Moriya interaction—were assigned Gaussian distributions to emulate realistic disorder.

We then examined SOT-driven domain-wall motion under varying current densities. The influence of pinning is reflected in the velocity–current density (v – J) characteristics (**Fig. R2b**). In both device sizes, we observe depinning and creep regimes—hallmarks of pinning-limited motion. The smaller device exhibits a higher depinning threshold current, indicating stronger pinning effects due to the reduced dimensions. Once the depinning transition is surpassed, the motion enters the flow regime, where the dynamics become largely independent of pinning [3]. This flow regime represents the desirable operational state for domain-wall motion devices.

In conclusion, **device downscaling increases the required current density to achieve a given domain-wall velocity.**

Fig. R2 | Influence of domain-wall pinning in downscaled devices. (ai), Domain wall motion device. Black: $-M_z$, White: $+M_z$, Red: Domain Wall region. **(aia)**, The device is divided into grains with Gaussian distributed size and material properties. Different color represents different perpendicular magnetic anisotropy values. **(b)**, Velocity vs. Current Density plot.

Reference

- [1] Moretti, S., Voto, M. and Martinez, E., 2017. Dynamical depinning of chiral domain walls. *Physical Review B*, 96(5), p.054433.
- [2] Herranen, T. and Laurson, L., 2019. Barkhausen noise from precessional domain wall motion. *Physical review letters*, 122(11), p.117205.
- [3] Metaxas, P.J., Jamet, J.P., Mougín, A., Cormier, M., Ferré, J., Baltz, V., Rodmacq, B., Dieny, B. and Stamps, R.L., 2007. Creep and Flow Regimes of Magnetic Domain-Wall Motion in Ultrathin Pt/Co/Pt Films with Perpendicular Anisotropy. *Physical review letters*, 99(21), p.217208.

Revisions:

This section is added into the Supplementary Fig. S4.

A description is added into the Main Text: From Line 191 to 192.

2. Though the velocity of current-driven domain-wall motion can be fast, the detection of Hall voltage is through DC measurement in this work, which slows down the operation speed. The author should demonstrate a fast way to read the Hall signal to match the fast domain-wall velocity. In addition, how is the domain-wall velocity in this work? They should measure the domain-wall velocity as a function of current density.

Reply:

We thank the reviewer for this valuable suggestion. We have developed a fast Hall measurement technique and experimentally determined the domain-wall velocity.

In our setup, the Hall voltage is recorded using a high-speed oscilloscope with a detection bandwidth in the MHz range, allowing real-time tracking of sub-10 m/s domain-wall motion. For higher velocities (on the order of hundreds of m/s), GHz-level detection would be required, which can be achieved by integrating a high-frequency detection circuit, as demonstrated in Reference [1]. Due to time constraints, this GHz-range implementation was not pursued in the current study but remains an extension of our setup.

The measurement principle is detailed as follows. It relies on the linear dependence of the Hall voltage (V) on the domain length (L_D) within the Hall channel. As illustrated in **Fig. R3a**, V is expressed as:

$$V = V_{AHE} \left(2 \frac{L_D}{L_P} - 1 \right) \quad (1)$$

where L_P is the Hall channel length, and V_{AHE} is the saturated anomalous Hall effect (AHE) voltage.

When a current drives the domain wall, L_D evolves with time as $L_D = v_{DW} \times t$, giving:

$$V = V_{AHE} \left(2 \frac{v_{DW} \cdot t}{L_P} - 1 \right) \quad (2)$$

Hence, the domain-wall velocity (v_{DW}) can be extracted from the slope of the $V - t$ curve.

Using this method, we measured the domain-wall velocity in our MCA device (**Fig. R3b**). The real-time Hall Voltage vs. Time curve is shown in **Fig. R3c** (right Y-axis), and the corresponding domain length (L_D) derived from Eq. (1) is plotted on the left Y-axis. Linear fitting of $L_D - t$ yields a slope of 0.0625 m/s, representing the domain-wall velocity. The dependence of velocity on current density is summarized in **Fig. R3d**.

Reference

[1] Sala, G., Krizakova, V., Grimaldi, E. et al. Real-time Hall-effect detection of current-induced magnetization dynamics in ferrimagnets. Nat Commun 12, 656 (2021).

Revisions:

This section is added into the Supplementary Fig. S1.

A description is added into the Main Text: From Line 179 to 181.

Fig. R3 | Domain wall motion speed measurement. (a), Blue rectangle represents the $-M_z$ domain, red the $+M_z$ domain, and grey the Hall channel. L_P : Hall channel length. L_D : Length of the $-M_z$ domain inside the Hall channel. A current j is applied to drive the domain wall motion. The voltage V of the Hall channel is measured in real time. Top left picture illustrates the ideal Voltage vs. Time plot for the domain wall motion. (b), Current-driven domain wall motion in the MCA device. Horizontal rectangle is the magnetic strip. Black color is the $-M_z$ domain, and grey the $+M_z$ domain. Pairs of vertical strips are Hall channels. (c), L_D and Hall Voltage vs. Time plot measured in the rightmost Hall channel. The velocity is obtained by linear fitting. (d), Domain-wall velocity as a function of current density.

Besides, some other issues need to be addressed:

1. The device used by the authors is based on the ferromagnet CoFeB. The velocity of ferromagnetic DW motion is usually limited to a few hundred meters per second. The authors should provide more experimental evidence to support the expected operating speed of THz. Moreover, the size of the domain wall in ferromagnets is usually tens of nanometers. The authors should provide more evidence to support the expected domain size of 1 nm.

Reply:

We thank the reviewer for raising this important point and apologize for the lack of clarity in our original description. We have revised the manuscript to explicitly state that the THz operation

speed and fast domain-wall motion refer to **ferrimagnetic** systems, not the CoFeB-based ferromagnetic device used in this work. In addition, we clarified that the domain size discussed in the manuscript is **sub-10 nm**, rather than 1 nm, and is achievable in systems with large Dzyaloshinskii–Moriya interaction (DMI).

The revisions are shown below for your review; related references are also updated.

From Line 79 to 85 of the Main Text:

The system performance can be further enhanced by selecting alternative material systems. For example, systems with large Dzyaloshinskii–Moriya interaction offer exceptional scalability (domain sizes down to **sub-10 nm²¹**); low-roughness systems provide significantly higher energy efficiency (domain-wall motion energy ~ 27 aJ²²); and heavy-metal/**ferrimagnet** systems enable much higher operation speeds (domain-wall velocities exceeding 1000 m/s²³ and dynamic frequencies reaching the THz regime²⁴). Importantly, all these systems retain the key advantage of nonvolatility²⁵.

From Line 427 to 430 of the Main Text:

Furthermore, **leveraging diverse material systems** could enable scaling domain sizes down to **sub-10 nm²¹**, maintaining high on–off ratios³⁴, and boosting operational speeds up to ~ 1 THz²⁴. Pinning optimization could substantially reduce the domain wall driving energy to ~ 27 aJ²².

THz speed

Ferrimagnet exhibit **THz-range** dynamics. Several experimental studies are shown below:

Material	Switching Speed	Reference
GdFeCo	1.5 picosecond	[1]
Pt/Co/Gd	100 femtosecond	[2]
Mn ₂ Ru _x Ga	2 picosecond	[3]

In these systems, typical domain-wall motion velocities **exceed 1000 m/s**, as summarized by experimental results below:

Material	Max Speed	Driving Method	Reference
GdFeCo	1500 m/s	Magnetic field	[4]
Pt/GdFeCo	1300 m/s	Spin-orbit torque	[5]
Pt/BiYIG	Over 4300 m/s	Spin-orbit torque	[6]

The Pt/GdFeCo system can serve as an alternative to our Ta/CoFeB/MgO structure, as it provides a large anomalous Hall signal (Convolution Kernel and Reading) [7,8], supports perpendicular multidomain states (Convolution Input), and enables spin–orbit-torque-driven domain-wall motion (Convolution Shifting) [5], fulfilling the essential requirements for convolution operation. Therefore, in the revised manuscript, we describe this as an alternative system.

Sub-10 nm domain size

Systems with large DMI can support small domain sizes. As shown in **Fig. R4** (adapted from Reference [3]), the combined domain and domain-wall region (circled in red) measures

approximately $5 \times 2 \text{ nm}^2$. Accordingly, we have revised the manuscript to clarify that the domain size is **sub-10 nm**, rather than 1 nm.

This system, consisting of a Pd/Fe bilayer on an Ir(111) single crystal, can also serve as an alternative to our Ta/CoFeB/MgO structure, as both fulfill the essential requirements for convolution operation. Therefore, in the revised manuscript, we refer to this as an alternative system.

[FIGURE REDACTED]

Fig. R4 | Sub-10 nm domain. Spin-polarized scanning tunneling microscopy image of a Pd/Fe island on Ir(111) showing the multi-domain state. Differential conductance (dI/dU) map gives the magnetic contrast of the out-of-plane (z) direction. Figures adapted from Reference [9].

Reference

1. Radu, I., Vahaplar, K., Stamm, C., Kachel, T., Pontius, N., Dürr, H.A., Ostler, T.A., Barker, J., Evans, R.F.L., Chantrell, R.W. and Tsukamoto, A., 2011. Transient ferromagnetic-like state mediating ultrafast reversal of antiferromagnetically coupled spins. *Nature*, 472(7342), pp.205-208.
2. Lalieu, M.L.M., Peeters, M.J.G., Haenen, S.R.R., Lavrijsen, R. and Koopmans, B., 2017. Deterministic all-optical switching of synthetic ferrimagnets using single femtosecond laser pulses. *Physical review B*, 96(22), p.220411.
3. Banerjee, C., Teichert, N., Siewierska, K.E., Gercsi, Z., Atcheson, G.Y.P., Stamenov, P., Rode, K., Coey, J.M.D. and Besbas, J., 2020. Single pulse all-optical toggle switching of magnetization without gadolinium in the ferrimagnet Mn₂RuGa. *Nature communications*, 11(1), p.4444.
4. Kim, Kab-Jin, et al. "Fast domain wall motion in the vicinity of the angular momentum compensation temperature of ferrimagnets." *Nature materials* 16.12 (2017): 1187-1192.
5. Caretta, Lucas, et al. "Fast current-driven domain walls and small skyrmions in a compensated ferrimagnet." *Nature nanotechnology* 13.12 (2018): 1154-1160.
6. Caretta, L., Oh, S.H., Fakhrul, T., Lee, D.K., Lee, B.H., Kim, S.K., Ross, C.A., Lee, K.J. and Beach, G.S., 2020. Relativistic kinematics of a magnetic soliton. *Science*, 370(6523), pp.1438-1442.
7. Roschewsky, N., Matsumura, T., Cheema, S., Hellman, F., Kato, T., Iwata, S. and Salahuddin, S., 2016. Spin-orbit torques in ferrimagnetic GdFeCo alloys. *Applied Physics Letters*, 109(11).
8. Seung Ham, W., Kim, S., Kim, D.H., Kim, K.J., Okuno, T., Yoshikawa, H., Tsukamoto, A., Moriyama, T. and Ono, T., 2017. Temperature dependence of spin-orbit effective fields in Pt/GdFeCo bilayers. *Applied Physics Letters*, 110(24).
9. Spethmann, J., Vedmedenko, E.Y., Wiesendanger, R., Kubetzka, A. and von Bergmann, K., 2022. Zero-field skyrmionic states and in-field edge-skyrmions induced by boundary tuning. *Communications Physics*, 5(1), p.19.
10. Romming, N., Kubetzka, A., Hanneken, C., von Bergmann, K. and Wiesendanger, R., 2015. Field-dependent size and shape of single magnetic skyrmions. *Physical review letters*, 114(17), p.177203.

2. They applied a high current to displace the domain wall, which may induce significant Joule heating, particularly when operating at a high frequency. The authors should consider this in their modelling and estimate the device temperature due to Joule heating.

Reply:

We thank the reviewer for this valuable suggestion. We performed additional experiments to characterize the heating effect.

First, we measured the R_{xx} - T curve (device longitudinal resistance vs. temperature) for the same devices used in the manuscript. The setup is shown in **Fig. R5a**. A 1 μA reading current (chosen to minimize self-heating) was applied to the current-shifting channel, and the voltage was measured across the two Hall channels using the standard four-probe method. The resulting R_{xx} - T curve (**Fig. R5b**) shows a resistance of 349.72 Ω at 300 K with a slope of $-0.069 \Omega/\text{K}$, which serves as our calibration reference. The temperature rise (ΔT) associated with Joule heating during domain shifting can be obtained from the measured change in longitudinal resistance (ΔR_{xx}) using: $\Delta T = -\frac{1\text{K}}{0.069\Omega} \Delta R_{xx}$.

To quantify the heating effect, we applied DC currents ranging from 1 μA to 1 mA and measured the equilibrium R_{xx} (**Fig. R5c**). For a 1 mA current—the shifting current used in the manuscript—we obtain: $\Delta T = \frac{1\text{K}}{0.069\Omega} \Delta R_{xx} \rightarrow \Delta T = -\frac{1\text{K}}{0.069\Omega} \times (348.33\Omega - 349.72\Omega) \approx 20 \text{ K}$. This indicates that the device temperature increases from **300 K to approximately 320 K during 1mA DC current application**.

A temperature rise of $\sim 20 \text{ K}$ is modest in magnetic device applications and is unlikely to cause irreversible changes in magnetic properties. Larger temperature change is commonly observed in spin-orbit torque and domain-wall motion studies, without performance degradation.

Moreover, in practical operation, the device is driven by pulse with a 50% duty cycle (or lower), which further reduces the effective heating. Therefore, Joule heating is not expected to notably affect the reliability or reproducibility of our device.

Revisions:

This section is added into the Supplementary Fig. S2.

A description is added into the Main Text: From Line 179 to 181.

Fig. R5 | Heating effect during Shifting operation. (a), Measurement setup. **(b),** R_{xx} vs. T curve. **(c),** R_{xx} as a function of DC current amplitude. Each current is applied for 4 hours, and measurements are performed at room temperature.

3. There have been several reports using physical systems for convolution computing. The author should compare with other systems and clarify the advantages of MCA devices.

Reply:

We thank the reviewer for this valuable question. We identify the multiply-and-accumulate (MAC) function as the fundamental operation in physical convolution computing. Based on this, we conducted a literature survey and identified relevant optical implementations for benchmarking. The comparative results are summarized below:

Benchmarks	Our Device	Ref. [1]	Ref. [2]
Energy	4.75fJ to 1.01pJ per MAC	0.53pJ per MAC	25fJ per MAC
Latency	4.03ns	1ns	0.25ps
Area	133um ² to 103um ²	1cm ²	1mm ²

Benchmarking energies are per MAC operation. Accordingly, we used the values from Main Text Fig. 4b (4.86pJ and 1.03nJ) and divided them by 1024, as they are for 1024-point convolution.

Compared with Refs. [1,2], our device exhibits comparable energy efficiency and higher latency but offers a dramatically smaller area footprint, which is critical for large-scale on-chip integration.

It is also important to note that the estimations of photonic convolution systems (Refs. [1,2]) are typically based on device-level metrics and do not include system-level overheads such as optical-to-electrical conversion via photodiodes, which introduce additional latency and energy consumption.

In contrast, our benchmarking accounts for the energy/latency/area cost from all peripheral and interfacing components, including CMOS integration for control, ADCs, and post-processing. Specifically, we include the contributions from voltage drivers, switching capacitors, ADCs, and both electric–magnetic and magnetic–electric conversion lines, providing a more comprehensive and realistic estimate of practical system performance. More details can be found in **Comment 10** from **Reviewer 2**.

At present, we have identified only these two representative physical convolution studies. We remain open to benchmarking against any other existing physical convolution systems if further comparisons are requested.

Reference

[1] Amiri, S. and Miri, M., 2025. All-optical convolutional neural network based on phase change materials in silicon photonics platform. *Scientific Reports*, 15(1), p.22055.

[2] Fan, L., Wang, K., Wang, H., Dutt, A. and Fan, S., 2023. Experimental realization of convolution processing in photonic synthetic frequency dimensions. Science Advances, 9(32), p.eadi4956.

Revisions:

A discussion is added in the Main Text from Line 398 to 401; the references are also updated:

Compared with optical systems for physical-convolution computing^{32,33}, the MCA exhibits a comparable level of energy consumption but higher latency, primarily due to the lower speed of domain-wall motion. Nonetheless, its area footprint is orders of magnitude smaller, offering clear advantages for large-scale integration and on-chip deployment.

4. The convolutional algorithm relies on the fact that the Hall voltage is linear with respect to the length of domains, whereas there are nonlinear deviations in the real devices (Fig. 2f and 2h). The author should analyze how this affects the algorithm.

Reply:

We thank the reviewer for raising the important question regarding the effect of nonlinearity on the convolution algorithm. To address this, we use handwritten digit recognition as a representative example and perform experiments using the device's raw data that inherently include the nonlinear deviations. The response is divided into two parts:

1. Nonlinear deviations from experiment
2. Algorithm performance with nonlinear deviations included

Nonlinear deviations from experiment

In our experimental setup, the nonlinear deviations mainly come from two sources: (1) variations in the domain length (input variation) and (2) variations in the Hall pad width (weight variation).

The nonlinear deviations of these two factors are presented in Main Text Figs. 2f and 2h, and reproduced in **Fig. R6** for reference. Quantitatively, the average deviation in Hall pad width is approximately 7%, while the variation in domain length is about 2%.

Algorithm performance with nonlinear deviations included

(a) Input variation

We compared the input data using the raw experimental results from Fig. 2h with the ideal case. As shown in **Fig. R7**, the handwritten digit datasets with and without domain-length deviations are visually indistinguishable, as the experimental variation in domain length is only about **2%**. This small deviation preserves the fundamental image features necessary for accurate handwritten-digit recognition. Consequently, the recognition accuracy remains **98.94%**, demonstrating that input nonlinearity has a negligible effect on the algorithm’s performance.

Fig. R7 | Input with and without deviation. Left: MNIST image without incorporating domain variation. **Right:** MNIST image with domain length variation. Both images show a clear digit shape and the difference is negligible.

(b) Weight variation

In the original manuscript, we already accounted for variations in the weights by directly using the raw V_{AHE} readouts from different Hall channels as the weight values.

As discussed in the manuscript, the training process was performed in software, whereas the MCA device is designed for inference with fixed weights. Accordingly, the kernel values derived from the training results are approximated using the nearest experimental V_{AHE} values, as illustrated below:

Ideal convolution kernel (training):

0.1399	-0.289	0.172
0.489	0.251	-0.521
0.2173	-0.299	-0.452

Raw V_{AHE} approximation (experiment):

0.1400	-0.271	0.184
0.490	0.239	-0.524
0.2208	-0.305	-0.478

We then performed inference using both **experimental input** and **weight variations** (raw V_{AHE} values incorporating domain-length and Hall-pad deviations). The resulting recognition accuracy remains high at **98.80%**, confirming that the observed nonlinearities do not significantly impact the algorithm’s performance.

Revisions:

This section is added into the Supplementary Fig. S10.

A description is added into the Main Text: From Line 327 to 330.

5. The authors designed several MCA devices for different kernel functionalities. What is the principle for designing the MCA devices? How does the influence of the Hall probe number?

Reply:

We thank the reviewer for asking about the design principle of the MCA devices.

The design of the MCA device is fundamentally based on the convolution operation. Any functionality that can be expressed by a convolution kernel can, in principle, be realized through an MCA by tailoring the device geometry accordingly. In this framework, the convolution input is encoded by the domain lengths, while the kernel is defined by the spacing (W_p) and number of the Hall probes, which determine the convolution weights.

For example, in the case of a Gaussian blurring kernel commonly used in image processing, the kernel coefficients follow a Gaussian distribution. We translate this distribution into Hall-pad spacing, using a sequence of $W_p = 3-12-18-12-3 \mu\text{m}$, which approximates the Gaussian curve. This design employs five Hall probes, as shown in **Fig. R8** (left). Increasing the number of Hall probes (e.g., to seven or nine) improves the resolution of the Gaussian approximation.

Similarly, for edge detection, the kernel follows the function $(1, 0, -1)$, which measures the intensity difference between neighboring pixels (see **Comment 6** for details). The corresponding device uses $W_p = 8-8-8 \mu\text{m}$, reflecting the equal magnitude of the coefficients $+1$ and -1 . The opposite signs are implemented by reversing the connection polarity of the first and third Hall channels, while the 0 coefficient is realized by leaving the middle Hall channel unconnected. Although **Fig. R8** (right) shows five equally spaced Hall pads ($8-8-8-8-8 \mu\text{m}$), only three are active for edge detection, while all five are utilized for the averaging function.

In summary, the design principle of the MCA device is determined by the target convolution kernel, where the number and spacing of Hall probes (W_p) directly correspond to the kernel's elements and their relative magnitudes. Further details are provided in the **Methods** section of the manuscript.

6. It is not clear to me how they perform the handwritten digit recognition and image edge detection tasks. Which results were performed in physical devices and which results were performed in computers? They should provide more details, such as how to encode the grey scale value into domain length, how to choose the current density and how to correct the offset Hall voltage.

Reply:

We thank the reviewer for asking about the detailed implementation of the MCA device. We realize that our original description may not have been sufficiently clear. The details are clarified below in the following order:

1. Encoding grey scale value, Choice of current density, and Correction of offset Hall voltage
2. Image edge detection
3. Handwritten digit recognition

Encoding grey scale values

As shown in **Fig. R9a**, each image is composed of pixels with grayscale values ranging from 0 to 255. An example of pixel grayscale variation from pixel No. 100 to No. 150 is shown in **Fig. R9b**. The corresponding V_{AHE} -domain-length dependence is measured experimentally (**Fig. R9c**), establishing a one-to-one linear mapping between grayscale value and domain length through V_{AHE} : Grey scale (Computer) \rightarrow V_{AHE} (Device) \rightarrow Domain length. This mapping allows each pixel's intensity to be physically represented by a domain of proportional length in the MCA device.

In summary, the grayscale values (from the original image) are obtained in the computer, while the corresponding V_{AHE} values are measured experimentally in the MCA device.

[FIGURE REDACTED]

Fig. R9 | Encoding grey scale value to domain length. (a), Grey scale image. **(b)**, Grey scale value of pixel No. 100 to No. 150. **(c)**, V_{AHE} as a function of domain length.

Choice of current density

In our current implementation, a fixed driving current of 1 mA is used for domain-wall motion, corresponding to a current density of 8.33×10^9 A/m². The choice of current density is determined by the device geometry and the temporal resolution of the pulse generator. Since the pulser used in our experiments provides reliable control in the μs -range, we selected a domain-wall velocity of approximately 0.5 m/s, corresponding to a $40\mu\text{s}$ displacement time for a $20\mu\text{m}$ Hall-electrode spacing.

Correction of offset Hall voltage

The offset Hall voltage arises primarily from slight misalignment of Hall pads during fabrication (see also **Comment 8, Reviewer 3**). At the current stage, this offset is corrected computationally in computer. Experimentally, it can be minimized using high-resolution lithography, as demonstrated in **Fig. R10**, where the offset is nearly eliminated.

Alternatively, offset compensation can be achieved using a simple peripheral differential amplifier or offset-cancellation circuit. The offset can be effectively subtracted in real time using standard readout electronics commonly employed in Hall sensors and magnetic memory devices [1,2]. Such circuits require only a modest number of transistors and consume microwatt-level power, negligible compared with the total device array, while operating well within the readout bandwidth of magnetic devices [3].

Fig. R10 | AHE measurement of device with improved fabrication. The alignment is improved and the offset is nearly zero.

Reference

- [1] Ramsden, E., 2011. Hall-effect sensors: theory and application. Elsevier.
- [2] Popovic, R.S., 2003. Hall effect devices. CRC Press.
- [3] Carusone, T.C., Johns, D.A. and Martin, K.W., 2011. Analog integrated circuit design. John Wiley & Sons.

Image edge detection

We first briefly discuss the principle of edge detection, as illustrated in **Fig. R11a**. Consider a one-dimensional line of pixels, where the edge corresponds to the region with the largest change in grayscale value—for example, between 1 and -1 in the sequence. In flat regions (e.g., $1\ 1\ 1$ or $-1\ -1\ -1$), no change occurs. To detect the edge, we use a convolution kernel of $(1, 0, -1)$ and slide it along the pixel line. In flat regions, the convolution output remains 0, while at the edge, it becomes nonzero (2 in this example). Consequently, only the edge appears bright (grayscale = 2), and all other regions remain dark (grayscale = 0).

Thus, the edge-detection process follows the standard convolution operation, consisting of shift and multiplication steps. The shift is achieved through SOT-driven domain-wall motion, while multiplication is realized by combining the input and kernel values. The input grayscale image is encoded as the domain length (L_D), and the kernel values are encoded as the spacing between pairs of Hall pads (W_P). These two parameters produce a Hall voltage proportional to $V_{AHE} \sim L_D \times W_P$. The resulting multiplication outputs (V_{AHE}) are then summed in the computer to reconstruct the edge-detected image. The overall flow is shown in **Fig. R11b**. In the future, this summation can be fully implemented in hardware using a simple switched-capacitor circuit, as illustrated in **Fig. R11c**, enabling a completely integrated operation.

Fig. R11 | Edge detection. (a), Principle. (b), Flow diagram. (c), Switch capacitor circuit.

Handwritten digit recognition

The handwritten-digit recognition task is implemented using a convolutional neural network, as shown in **Fig. R12a**. Only the convolutional layers (Conv) are physically realized using the MCA device, while the subsequent layers—max pooling, flattening, and fully connected layers—are executed on a computer. As discussed in the manuscript, convolution layers account for over 90% of the total computation cost; thus, implementing them in our MCA hardware significantly reduces energy consumption and area footprint.

The physical implementation of the convolution layer follows the same principle as the edge-detection operation, differing only in the kernel size and values. For instance, the grayscale image matrix is convolved with a 3×3 kernel (**Fig. R12b**). Each 3×3 block of the image is multiplied element-wise by the kernel matrix and summed to yield the first element of the output feature map. The kernel is then shifted across the entire image matrix to complete the convolution.

Similar to every convolution operation performed by our MCA device, here the pixel value (input) is encoded as domain length, the kernel (weight) is encoded as the spacing between Hall pads (**Fig. R12c**), and the shift and multiplication are done physically.

Fig. R12 | Handwritten digit recognition. (a), Flow diagram. MCA device is used to reduce the energy and area of the Conv layers. **(b)**, Principle of the convolution layer. **(c)**, Physical implementation of convolution.

Reviewer #2 (Remarks to the Author):

Authors proposed a nice idea having the weights mapped as spacing between electrode pairs during fabrication, which mean weights are fixed at that stage. The inputs, however are applied as domain lengths that can be dynamically controlled and changed at runtime. However, only this part of the device has been experimentally validated. At this stage, the work has multiple practical issues, which are not resolved or discussed. Furthermore, there are theoretical ideas which are not validated in depth, such as, 2D version where both the weights and inputs can be dynamically programmed.

Reply:

We sincerely thank the reviewer for the thorough and constructive feedback, which has greatly improved the practical relevance, scientific rigor, and overall impact of our work. In response, we have performed additional experiments, calculations, and algorithmic validations to address the reviewer's concerns and to further substantiate both the experimentally demonstrated and theoretically proposed aspects of the device. Moreover, we have substantially revised the Main Text and Supplementary Information to incorporate these new results and enhance the overall clarity and coherence of the manuscript

A detailed point-by-point response to each comment is provided below:

The critical points are highlighted below.

1. The manuscript acknowledges that the current 1D Hall-based MCA requires relatively high current densities to drive domain wall motion, leading to energy per operation that is slightly higher than CMOS. While the authors suggest this can be mitigated by material optimization, this is left as future work.

Reply:

We thank the reviewer for this insightful comment. In response, we conducted additional experiments to reduce the energy required for domain-wall motion through material optimization.

Our material stack is Ta (5) / CoFeB (1) / MgO (2), with thicknesses in nanometers. Motivated by Reference [1], we optimized the growth pressure of the heavy-metal Ta layer, which strongly influences defect density and interfacial quality. Specifically, we prepared three batches of samples with Ta growth pressures of 2.0, 2.5, and 3.0 mTorr and characterized their domain-wall motion velocities. As reported in Reference [1], a lower growth pressure typically reduces pinning-site density, thereby decreasing the energy required for domain-wall motion. Our measurements confirm a similar trend, as shown in **Fig. R1a**.

For the 2.0mTorr condition, a domain-wall velocity of 2.5 m/s is achieved under a current density of 2.1×10^9 A/m². In contrast, before optimization, the same velocity required a current density of approximately 11.5×10^9 A/m² (**Fig. R1b**). This corresponds to an **~30-fold reduction in energy consumption** per operation.

Fig. R1 | Domain wall motion after pinning optimization. (ai), Snapshots of domain wall motion. White (grey) color denotes +Mz (-Mz). V denotes domain wall motion velocity, and $t_{0,1,2}$ represent time stamps. **(a(ii),** Domain wall motion velocity vs. Current density plot. **(b),** Domain wall motion velocity vs. Current density plot before material optimization.

Reference

[1] Kumar, Durgesh, et al. "Ultralow energy domain wall device for spin-based neuromorphic computing." ACS nano 17.7 (2023): 6261-6274.

2. The device relies on precise domain nucleation, length control, and synchronized wall motion. The paper shows impressive lab-scale control but does not address large-array scaling. Scalability and uniformity are critical for any practical accelerator.

Reply:

We thank the reviewer for this insightful feedback. The scalability and uniformity concerns are indeed critical for practical accelerator implementation and are collectively addressed through our detailed responses to other specific comments.

The **scalability** issue is addressed by:

1. Domain wall synchronization measurement over large distance and methods for handling crosstalk or variation in large arrays (**Comment 6**)
2. Method for handling alignment, yield, and TMR variation across large arrays (**Comment 9**)

The **uniformity** issue is addressed by:

1. Quantitative measurements of deviation from ideal case (**Comment 4**)
2. Measured yield of fabrication (**Comment 5**)
3. Method for handling alignment, yield, and TMR variation across large arrays (**Comment 9**)

3. While a 2D MTJ-MCA extension is proposed, it is theoretical. Can the authors clarify whether any test structures for the 2D MTJ-MCA have been fabricated?

Reply:

We thank the reviewer for this valuable comment. In response, we clarify that we have fabricated test structures for the 2D MTJ-MCA and experimentally verified all key convolutional operations—namely Read, Write, Shift, and Tunable Weight. The results are presented in two parts:

1. Fabrication of MTJ-MCA devices on an industry-grade full-MTJ material stack (8-inch wafer)
2. Experimental demonstration of Read, Write, Shift, and Tunable Weight operations

Fabrication of MTJ-MCA devices on an industry-grade full-MTJ material stack (8-inch wafer)

In collaboration with a semiconductor foundry, we developed an industry-grade full-MTJ material stack on an 8-inch wafer. The wafer photograph and corresponding material stack are shown in **Fig. R2a** and **Fig. R2b**, respectively. Using this stack, we fabricated test MTJ-MCA devices, as shown in the optical micrograph in **Fig. R2c**. Each device consists of a magnetic strip (the Shifting Channel), a MTJ readout region (the Reading Junction, highlighted by the red dashed box), and two Oersted-field writing lines (the Writing Channel). The schematic on the right of **Fig. R2c** illustrates the detailed structure within the red dashed box.

Fig. R2 | Industry-grade full-MTJ material stack 8-inch wafer and MTJ-MCA device. (a), A photo of the 8-inch wafer. (b), Magnetic tunneling junction material stack. The layer thicknesses are not provided for confidential reasons. (c), MTJ-MCA device. The structure inside the red dashed box is illustrated in the right schematic. Red: -Mz domain. Blue: +Mz domain.

Experimental demonstration of Read, Write, Shift, and Tunable Weight operations

(a) Write:

Writing is performed using the Oersted field, same method as the Hall-MCA device. Representative MOKE snapshots are shown in **Fig. R3a**.

(b) Read:

Reading is performed using tunneling magnetoresistance (TMR), which, analogous to the anomalous Hall effect, is proportional to the net Mz. The measured TMR loop of the MTJ-MCA device is shown in **Fig. R3b**. The zoomed-in region (right panel) reveals distinct resistance steps, each corresponding to a convolution input value, with several hundred inputs achieved across the full loop.

(c) Shift:

Shifting is realized through spin-orbit torque (SOT)-driven domain-wall motion, employing the same mechanism as in the Hall-MCA. MOKE snapshots are provided in **Fig. R3c**.

Fig. R3 | Demonstration of Read, Write, and Shift and tunable weight in the MTJ-MCA device. (a), MOKE images of the Domain Writing process. Green shade: Oersted field channel. Yellow arrow: Applied current direction. A current is applied, and a +z domain is created, location mark by the blue arrow. (b), TMR loop. Each step corresponds to a convolution input. Right figure: Zoom-in plot of the red box region. (c), MOKE images of domain wall motion. Yellow shade: Reading junction. t1 to t6 represent the time stamps. Black arrow denotes the motion direction. **d(i)**, TMR loops by applying different junction voltages. Weight is obtained by $(\frac{R_{ap}-R_p}{R_p})$. **d(ii)**, Plot summarizing the measured R_p and R_{ap} at different applied voltages. Inset: weight values $(\frac{R_{ap}-R_p}{R_p})$. Currently, 15 different weights are obtained.

(d) Tunable weights:

In the Hall-MCA, the convolution weights are fixed by the geometric width between pairs of Hall pads. In contrast, the MTJ-MCA enables tunable weights, achieved by modulating the TMR ratio, defined as $\frac{R_{ap}-R_p}{R_p}$, where R_p (R_{ap}) denote the resistances corresponding to parallel and antiparallel alignments of the top and bottom CoFeB layers, respectively.

By applying different junction voltages, the TMR loops can be rescaled. As shown in **Fig. R3d(i)**, each junction voltage produces a distinct TMR loop that differs only in its TMR ratio. In other words, these loops can be transformed into one another by a multiplicative factor—corresponding to the definition of a weight.

As discussed in the **(b) Read** section, each point on the TMR loop represents a convolution input value. By adjusting the TMR ratio, the multiplicative factor (weight) of each input can thus be tuned. For instance, Point 1 at 50 mV transforms into Point 2 at 500 mV, effectively multiplying the input by a scaling factor (weight). Measurements under multiple junction voltages yielded a wide range of tunable weights, summarized in **Fig. R3d(ii)**.

The underlying mechanism is that increasing the junction voltage (and thus the electric field) allows deeper, un-spin-polarized electronic states to contribute to tunneling, thereby reducing the TMR ratio. We note that this voltage-controlled method is simpler than the originally proposed scheme in the Main Text, where the weights are tuned by varying the magnetization of the top CoFeB layer. However, the voltage-controlled range is limited, as higher voltages may induce barrier breakdown. Therefore, the magnetization-based approach remains the preferred and more scalable method for future implementations.

In summary, we have experimentally verified the MTJ-MCA architecture.

4. AHE voltage readout relies on linear mapping of domain length and electrode spacing, but fabrication imperfections introduce nonlinearity. Readout fidelity is fundamental to inference accuracy. Can the authors provide quantitative measurements of deviation from ideal linearity? How does readout noise impact classification accuracy in their CNN demonstration?

Reply:

We thank the reviewer for this valuable suggestion. Our response is divided into two sections:

1. Quantitative measurements of deviation from linearity
2. Readout noise and classification accuracy of CNN demonstration

Quantitative measurements of deviation from linearity

In our experimental setup, the nonlinear deviations mainly come from two sources: (1) variations in the domain length (input variation) and (2) variations in the Hall pad width (weight variation).

The nonlinear deviations of these two factors are presented in Main Text Figs. 2f and 2h, and reproduced in **Fig. R4** for reference. Quantitatively, the average deviation in Hall pad width is approximately **7%**, while the variation in domain length is about **2%**.

Fig. R4 | Main Text Fig. 2f & 2h. Left figure: Fig. 2f, weight variation. **Right figure:** Fig. 2h, input variation.

Readout noise and classification accuracy of CNN demonstration

(a) Domain length (input) variation

We compared the input data using the raw experimental results from Fig. 2h with the ideal case. As shown in **Fig. R5**, the handwritten digit datasets with and without domain-length deviations are visually indistinguishable, as the experimental variation in domain length is only about **2%**. This small deviation preserves the fundamental image features necessary for accurate handwritten-digit recognition. Consequently, the recognition accuracy remains **98.94%**, demonstrating that input nonlinearity has a negligible effect on the algorithm's performance.

Fig. R5 | Input with and without deviation. Left: MNIST image without incorporating domain length variation. Right: MNIST image with domain length variation. Both images show a clear digit shape and the difference is negligible.

(b) Hall electrode spacing (weight) variation

In the original manuscript, we already accounted for variations in the weights by directly using the raw V_{AHE} readouts from different Hall channels as the weight values.

As discussed in the manuscript, the training process was performed in software, whereas the MCA device is designed for inference with fixed weights. Accordingly, the kernel values derived from the training results are approximated using the nearest experimental V_{AHE} values, as illustrated below:

Ideal convolution kernel (training):

0.1399	-0.289	0.172
0.489	0.251	-0.521
0.2173	-0.299	-0.452

Raw V_{AHE} approximation (experiment):

0.1400	-0.271	0.184
0.490	0.239	-0.524
0.2208	-0.305	-0.478

We then performed inference using both **experimental input** and **weight variations** (raw V_{AHE} values incorporating domain-length and Hall-pad deviations). The resulting recognition accuracy remains high at **98.80%**, confirming that the observed nonlinearities do not significantly impact the algorithm's performance.

Revisions:

This section is added into the Supplementary Fig. S10, highlighted in yellow.

A description is added into the Main Text: From Line 327 to 330, highlighted in yellow.

5. The device requires precise deposition of Ta/CoFeB/MgO stacks with PMA, fine Hall electrode spacing, and integration of Oersted nucleation lines. Scalable production is necessary for any hardware accelerator. What is the measured yield of fabricated devices? How sensitive is performance to lithographic variations?

Reply:

We thank the reviewer for this important question regarding device yield and sensitivity to lithographic variations. Our response is divided into two sections:

1. Measured device yield
2. Performance sensitivity to lithographic variations

Measured device yield

An optical image of the fabricated sample is shown in **Fig. R6a**, containing 354 MCA devices. A magnified view of the top block and a single representative device are shown in **Figs. R6b** and **R6c**, respectively.

To qualify as functional, a device must successfully perform the Write, Shift, and Read operations. Across the entire project, we have measured 64 MCA devices, of which 58 were fully operational, corresponding to a yield of **~90.6%**, primarily limited by the Shift operation.

Fig. R6 | Sample and device. (a), An actual fabricated sample. **(b)**, A block of the sample. **(c)**, A single MCA device from (b).

(a) Write

Domain nucleation and length control are achieved using an Oersted-field line, as described in the manuscript. The mechanism and fabrication requirements are both simple and robust. The writing lines are tens of micrometers in width, easily handled by standard lithography. No failures were observed in the Write operation, giving a **100% yield**.

(b) Shift

The main source of yield loss arises from local fabrication defects that hinder domain-wall motion. A representative defective device is shown in **Fig. R7a**, with the magnified region in **Fig. R7b** highlighting structural imperfections. Such defects reduce domain-wall velocity, requiring higher current densities to achieve the same speed and thereby increasing energy consumption.

For quantitative evaluation, we measured domain-wall velocities in all 64 devices under the same driving current density of $8.33 \times 10^9 \text{ A/m}^2$ (the value used in the manuscript). The results, summarized in **Fig. R7c**, show that typical devices exhibit velocities around 0.5 m/s, while defective ones fall below 0.4 m/s. We therefore define devices with $v < 0.4 \text{ m/s}$ as non-functional. The resulting yield is **58/64 (~90.6%)**.

Fig. R7 | Device yield from the shifting operation. (a), Microscope image of an MCA device with defects. **(b)**, Zoom-in image of **(a)** showing the shape defect in magnetic strip. **(c)**, Device variation in domain motion velocity.

(c) Read

The Read operation relies on detecting the anomalous Hall effect (AHE), which scales linearly with the width W_P . AHE sensing is highly robust, with only small deviations from ideal linearity due to fabrication non-idealities. As demonstrated in **Comment 4**, the readout quality remains sufficient for algorithmic implementation. Thus, the Read operation also shows a **100% yield**.

In summary, the domain-wall motion (Shift) stage represents the primary yield bottleneck. Among 64 tested devices, 58 functioned correctly, **yielding ~90.6% overall**.

Yield	Write Success Rate	Shift Success Rate	Read Success Rate
90.6%	100%	90.6%	100%

Revisions:

This section is added into the Supplementary Fig. S5 & S6.

A description is added into the Main Text: From Line 208 to 214.

Performance sensitivity to lithographic variations

The lithographic variation in our lab-scale fabrication is approximately $1\text{--}2\ \mu\text{m}$, primarily limited by the instrument resolution (for example, Karl Suss MA-6 contact aligner). These variations result in two types of imperfections: (i) shape defects, and (ii) pattern misalignment.

Shape defects mainly affect the Shift operation by introducing pinning sites that reduce domain-wall velocity, as discussed in the previous section. In contrast, pattern misalignment laterally offsets device features by about $1\ \mu\text{m}$, but has negligible impact on all three operations (Write, Shift, and Read). This insensitivity arises because we intentionally included $3\ \mu\text{m}$ design margins in the photomask layout, providing sufficient spatial tolerance.

An exemplary mask design is shown in **Fig. R8a**, where the $3\ \mu\text{m}$ margin is highlighted. A representative optical micrograph of a fabricated device is shown in **Fig. R8b**. From the observed dimensions, it is clear that a $1\ \mu\text{m}$ misalignment remains well within the designed tolerance and does not affect device functionality.

Fig. R8 | Exemplary Mask design and a typical device. (a), Design margin of the MCA mask. **(b),** Microscope image of a typical MCA device.

6. While small-scale devices are demonstrated, large-scale arrays needed for real AI workloads are not. Demonstrating a few cells is valuable, but deployment requires large arrays. How does domain wall synchronization degrade over distance? What strategies exist to handle crosstalk or variation in large arrays?

Reply:

We thank the reviewer for this valuable comment. Our response is divided into three sections:

1. Domain wall synchronization measurement over distance
2. Handling of crosstalk
3. Handling of variations

Domain wall synchronization measurement over distance

We used MOKE imaging to quantitatively assess domain-wall synchronization during propagation. As illustrated in **Fig. R9a**, a domain is bounded by two domain walls (DWs), and their spacing—the domain length (L_D)—may fluctuate during motion. Thus, the degree of synchronization can be evaluated by measuring the variation in L_D as the domain moves along the track.

We experimentally prepared two representative cases: a wide domain and a narrow domain. The result is shown in **Figs. R9b & c**. For wide-domain propagation, the measured domain lengths exhibit a standard deviation of **0.15 μm (1.9%)**, and for narrow domains, **0.10 μm (2.1%)**. These results confirm the high stability of domain-wall motion.

Revisions:

This section is added into the Supplementary Fig. S3.

A description is added into the Main Text: From Line 188 to 191.

Handing of crosstalk

In large arrays, crosstalk may occur when neighboring MCA devices are placed too close together. The primary source of crosstalk is the dipolar (stray) field generated by the magnetic layer of adjacent devices.

To quantify this effect, we calculated the stray-field distribution based on the actual device dimensions: the magnetic layer (Domain Channel) measures $200\ \mu\text{m} \times 20\ \mu\text{m} \times 1\ \text{nm}$, with a saturation magnetization $M_s=1000\ \text{emu/cc}$. The results are summarized in **Fig. R10**. **Fig. R10a** shows the device geometry, and **Fig. R10b** maps the out-of-plane field component (B_z) near the device. The detailed dependence of stray-field magnitude on distance from the device edge is presented in **Figs. R10c** and **R10d** (short and long edges, respectively), with zoomed-in views from 100 nm to 1000 nm shown in **Figs. R10e** and **R10f**.

The stray field magnitudes at characteristic lengths are summarized below:

- **At 10 nm from the device edge:**
 $|B_z| \approx 19\text{--}20\ \text{mT}$; the tangential in-plane component is $\approx 3\ \text{mT}$ (that's $|B_y|$ at a long edge or $|B_x|$ at a short edge)
- **At 100 nm:**
 $|B_z| \approx 2.0\ \text{mT}$; tangential in-plane $\approx 0.03\ \text{mT}$ ($30\ \mu\text{T}$).
- **At 1 μm :**
 $|B_z| \approx 0.19\ \text{mT}$; tangential in-plane $\approx 0.0003\ \text{mT}$ ($0.3\ \mu\text{T}$).

For comparison, the coercivity of our magnetic layer is $\sim 5\ \text{mT}$. Therefore, a $1\ \mu\text{m}$ lateral spacing ensures that the maximum stray field ($\sim 0.19\ \text{mT}$) remains well below the coercivity, effectively suppressing magnetic crosstalk between adjacent devices.

To further increase device density, 100 nm or smaller spacing can be adopted if the perpendicular magnetic anisotropy (PMA) is enhanced. This can be achieved, for example, by reducing the CoFeB thickness or by switching to material systems such as Pt/CoFeB/MgO or Pt/Co/AlO_x, which exhibit coercivities exceeding 100 mT. Under such conditions, stray-field interactions would be negligible.

We note that this estimation represents the worst-case crosstalk scenario, assuming a uniformly magnetized single-domain state. In practical devices, however, the magnetization alternates across domains, producing flux closure and substantial field cancellation. Consequently, the actual magnetic crosstalk in large arrays is expected to be significantly weaker than this upper bound.

Revisions:

This section is added into the Supplementary Fig. S15.

A description is added into the Main Text: From Line 401 to 403.

Fig. R10 | Stray field from the magnetic layer. (a), Device dimensions. **(b),** Stray field color map. **(c),** Stray field distribution from the short edge. **(d),** Stray field distribution from the long edge. **(e),** Zoom-in of (c) from 100nm to 1000nm. **(f),** Zoom-in of (d) from 100nm to 1000nm.

Handling of variations

As discussed in Comment 5, our current devices exhibit a yield of approximately 90.6%, primarily limited by the Shift operation. The yield loss arises from shape defects introduced during fabrication using lab-scale instruments, which impose limits on lithographic resolution and uniformity.

To mitigate these variations, two practical strategies can be employed:

1. **Foundry collaboration:** Partnering with a semiconductor foundry would significantly improve fabrication quality, as industry-standard tools provide much higher patterning precision and process yield. Given the simplicity of our device geometry, a 180 nm technology node is sufficient to realize large-scale arrays with minimal variation.

2. **High-resolution e-beam lithography:** Alternatively, employing electron-beam lithography—with achievable resolutions of 20 nm or finer—would effectively eliminate the edge roughness and shape defects responsible for domain-wall velocity variation.

Both approaches ensure high uniformity and reproducibility, thereby addressing the observed variability in domain-wall motion and enabling scalable array fabrication.

7. Can the authors clarify what CMOS baseline design was used? Have they benchmarked against state-of-the-art low-power MAC implementations?

Reply:

We thank the reviewer for this important question regarding the CMOS benchmarking baseline. We apologize for not clearly stating these details in the original manuscript. Our response is organized into two sections:

1. Details of the CMOS benchmarking baseline
2. Benchmarking against state-of-the-art Multiplication-Accumulation (MAC)

Details of the CMOS benchmarking baseline

To enable a fair comparison, a dedicated CMOS application-specific integrated circuit (ASIC) was designed to replicate the exact same functionality of our device—specifically, performing 8-bit, 1024-point convolutions. The architecture is illustrated in **Fig. R11**.

The CMOS baseline consists of:

- 8-bit shift registers for sequential input data shifting,
- 8-bit fixed-weight registers for storing convolution kernels,
- Parallel 8-bit multipliers to compute the input–weight products,
- An 8-bit adder tree to sum the partial products.

Fig. R11 | CMOS baseline. CMOS circuit architecture used for benchmarking. (This figure is a schematic of the design, the actual circuit layout is significantly larger and is not shown here for brevity.)

This is provided in Supplementary Fig. S14.

Benchmarking against the state-of-the-art Multiplication-Accumulation (MAC)

Each circuit component shown in Fig. R11 was synthesized using a standard digital cell library provided by the TSMC semiconductor foundry. The library offers state-of-the-art specifications for each cell, including physical layout information, delay characteristics, leakage properties, and energy consumption metrics, some exemplary specifications are shown in Fig. R12. These specifications ensure the **CMOS benchmark has a state-of-the-art low-power, high-speed, and small-area MAC implementation.**

The CMOS benchmarking was established as follows:

- **Area Estimation:** The reported area reflects the fully synthesized layout, accounting for both the active components and routing overheads.
- **Energy Estimation:** The reported energy consumption corresponds to the dynamic (active) energy during computation.
- **Throughput Estimation:** The path delay extracted from synthesis results was used to determine the operating throughput of the benchmark CMOS hardware.

Cell Information					
(Characterization Condition:Process=Fast-Fast-Global,Voltage=0.88v,Temp=0degreeC)					
PG Pin=VDD					
Cell Name	Gate Count	Width(um)	Leakage(nW)		
			Min.	Ave.	Max.
FA1D0BWP30P140	7	2.38	4.24478	5.58558	6.46222
FA1D1BWP30P140	7	2.38	4.44341	5.849	6.58825
FA1D2BWP30P140	8	2.66	7.02151	8.44445	9.19524
FA1D4BWP30P140	12.5	4.06	10.1931	15.24757	17.9073
FA1OPTCD1BWP30P140	8.5	2.8	4.55611	7.04596	8.40322
FA1OPTCD2BWP30P140	10.5	3.36	7.42744	10.16772	13.1241
FA1OPTSD1BWP30P140	10.5	3.5	5.78481	8.70781	9.45632
FA1OPTSD2BWP30P140	13	4.06	10.9208	11.89439	14.0121

Pin Description					
(Characterization Condition:Process=Fast-Fast-Global,Voltage=0.88v,Temp=0degreeC)					
Cell Name	Pin Cap.(pf)			Max Cap.(pf)	
	A	B	CI	CO	S
FA1D0BWP30P140	0.00173135	0.00160491	0.00121795	0.05947	0.05947
FA1D1BWP30P140	0.00195025	0.00185761	0.00127974	0.11893	0.11893
FA1D2BWP30P140	0.00193991	0.00185532	0.00128104	0.23787	0.23787
FA1D4BWP30P140	0.0033801	0.00348015	0.00217717	0.47573	0.47573
FA1OPTCD1BWP30P140	0.00264959	0.00242136	0.00183236	0.11893	0.11893
FA1OPTCD2BWP30P140	0.00315751	0.00291293	0.00183783	0.23787	0.23787
FA1OPTSD1BWP30P140	0.00332985	0.00311896	0.0025401	0.11893	0.11893
FA1OPTSD2BWP30P140	0.00382377	0.00360191	0.00300691	0.23787	0.23787

Propagation Delay(unit:ns)					
(Characterization Condition:Process=Fast-Fast-Global,Voltage=0.88v,Temp=0degreeC)					
Cell Name	Path	Parameter	Group1	Group2	Group3
			(<0.00116)pf	(0.00116-0.02937)pf	(>0.02937)pf
	A to CO	t _{PLH}	0.0293+6.2162*Cloud	0.0329+4.3*Cloud	0.0353+4.1272*Cloud
		t _{PHL}	0.0324+5.5699*Cloud	0.0356+3.6615*Cloud	0.0374+3.5491*Cloud

Fig. R12 | Standard digital cell library from TSMC. The metrics ensures the state-of-the-art low-power, high-speed, and small-area MAC performance.

8. The main application demonstrated is MNIST digit classification (28×28 images). A simple dataset can overstate practical readiness.

Reply:

We thank the reviewer for this insightful comment. To further validate the practical potential of our device, we extended our evaluation beyond the MNIST dataset to the more challenging CIFAR-10 image classification task. Compared with MNIST, CIFAR-10 images have a higher dimensionality ($32 \times 32 \times 3$) and involve convolutional layers with approximately $20\times$ more kernel weights, thereby providing a more stringent benchmark for computational performance.

In our experiments, we incorporated readout deviations originating from both domain length and weight variability. Representative CIFAR-10 images with and without domain-length deviation are shown in **Fig. R13a**, illustrating that the induced variation has a negligible visual impact—an important prerequisite for maintaining correct classification performance.

The classification results obtained from our MCA device are presented in **Fig. R13b**. The achieved recognition accuracy is approximately **80%**, which is comparable to state-of-the-art CIFAR-10 baselines (see Ref. [1], **Fig. R13c**). These results confirm that our MCA device sustains robust performance even for more complex datasets beyond MNIST.

We note that the current accuracy limit ($\sim 80\%$) primarily arises from the simplified neural network architecture used for this demonstration rather than any intrinsic device limitation. Further improvements can be expected by employing more advanced network structures—such as ResNet—which are beyond the scope of the present work.

[FIGURE REDACTED]

Fig. R13 | CIFAR-10. (a), CIFAR-10 image without and with domain length deviations. **(b)**, CIFAR-10 recognition matrix from our device. For example, when input is an Automobile image, we have 91.4% chance to recognize it as an Automobile. An average of ~80% accuracy is achieved. **(c)**, CIFAR-10 and MNIST accuracy of the state-of-the-art result. Adapted from Ref [1].

Reference

[1] Shridhar, K., Laumann, F. and Liwicki, M., 2019. A comprehensive guide to bayesian convolutional neural network with variational inference. *arXiv:1901.02731*.

Revisions:

This section is added into the Supplementary Fig. S9.

A description is added into the Main Text: From Line 325 to 327.

9. The 2D extension is presented analytically, but no experimental validation is shown. Proposing a 2D solution is important, but feasibility is unclear. What is the plan for fabricating test 2D MTJ cells? How will the authors address alignment, yield, and TMR variation across large junction arrays? What is the estimated energy/area/throughput of such 2D designs versus 1D Hall MCA?

Reply:

We thank the reviewer for this valuable comment. Our response is divided into three sections:

1. Fabrication plan for CMOS-integrated 2D MTJ cells
2. Addressing fabrication challenges in large arrays
3. Performance estimation of 2D versus 1D design

Fabrication plan for CMOS-integrated 2D MTJ cells

Our preliminary experimental validation of the 2D MTJ-MCA concept was presented in **Comment 3**, where we demonstrated all essential functions:

- Write: Oersted-field-based domain nucleation
- Shift: Spin-orbit torque (SOT)-driven domain-wall motion
- Read: Tunneling magnetoresistance (TMR)-based readout
- Tunable weights

Building on this foundation, our next steps—conducted in collaboration with the foundry—will focus on three progressive stages: (a) device scaling, (b) array fabrication, and (c) CMOS integration.

(a) Device scaling

The current MTJ-MCA devices are on the micrometer scale. We plan to downscale them using electron-beam lithography, which provides ~ 20 nm patterning resolution. The target MTJ junction diameter is 100 nm, corresponding to a geometric scaling factor of $100 \text{ nm} / 20 \text{ } \mu\text{m} \approx 0.005$. The layout of the scaled-down MTJ-MCA design is shown in **Fig. R14a**.

(b) Array fabrication

We have completed the first version of the mask and layout design for array-level testing. In collaboration with the foundry, we will fabricate 8×8 MTJ-MCA arrays, with the contact layout example shown in **Fig. R14b**. These arrays will serve to evaluate yield, uniformity, and inter-cell interference across multiple MTJ elements.

(c) CMOS integration

For CMOS integration, the MTJ material stack will be sputter-deposited on an 8-inch 28 nm CMOS wafer at room temperature. The critical MTJ junctions (100 nm diameter) will be

patterned using e-beam lithography. M1–M5 back-end-of-line (BEOL) interconnect processing will be implemented.

The integration architecture is as follows:

- The MTJ-MCA device will be built atop VIA56.
- The source line (SL) of each read/write/shift column connects to the MTJ’s fixed layer (top layer) through VIA56, bottom and top electrodes (BE, TECT, TE).
- The bit line (BL) connects to the MTJ free layer (bottom layer) through VIA56 and BE.
- On-chip routing is arranged as SL on M1, WL on M4, and BL on M5.

The full integration flow is illustrated in **Fig. R14c**.

In summary, the CMOS front-end and M1–M5 BEOL will be fabricated by the foundry, followed by MTJ stack deposition and patterning in our lab. The final product will be a functional CMOS-integrated MTJ-MCA chip capable of performing convolution operations.

Fig. R14 | Fabrication plan for scale-down, large-array, CMOS-integrated MTJ-MCA. (a), Device pattern of MCA with 100nm MTJ. **(b),** Large-array of MCA devices. The contact layout is shown here. **(c),** CMOS-integration flow.

Addressing fabrication challenges in large arrays

(a) Alignment variation

To ensure alignment accuracy during array fabrication, we will adopt e-beam lithography with high-precision alignment marks and employ step-and-repeat exposure to maintain <10 nm overlay accuracy across the array. For wafer-scale implementation, we will collaborate with a foundry to leverage advanced stepper lithography and automated overlay correction techniques already established in standard MRAM production lines, which routinely achieve alignment tolerances below 5 nm.

(b) Yield variation

Yield optimization will be addressed through redundant test structures and in-situ process calibration to monitor layer-thickness uniformity and etch-endpoint control across the wafer. Statistical process control (SPC) will be applied during sputtering, patterning, and lift-off to ensure uniformity of the magnetic and tunnel-barrier properties over large areas. As discussed in **Comment 5**, the primary yield bottleneck arises from device-shape variation, which affects domain motion. To mitigate this, special emphasis will be placed on maintaining sidewall smoothness through optimized ion-beam etching and hard-mask engineering. Uniformity in feature dimensions will be verified using high-resolution SEM and MFM imaging across representative dies. Additionally, redundant sub-arrays and isolated test cells will be included on each wafer to statistically test the yield, enabling iterative feedback for process refinement. These combined steps are expected to maintain domain-wall motion consistency and ensure high-yield fabrication across large arrays.

(c) TMR variation

Regarding TMR variation, we will employ post-deposition annealing and MgO-barrier thickness tuning to minimize junction-to-junction variability. The TMR uniformity achieved in state-of-the-art MTJ arrays fabricated with similar methods is typically within $\pm 5\text{--}10\%$, which is adequate for AI algorithm applications, where small resistance fluctuations can be tolerated.

In summary, lithographic alignment will be ensured through precision e-beam and stepper overlay; yield will be enhanced through process control, device-shape optimization, and redundancy; and TMR variation will be minimized through material and thermal optimization consistent with foundry-qualified MRAM standards.

Performance estimation of 2D versus 1D design

The performance estimation for the 2D design is summarized in Main Text Fig. 4b, with full details provided in the Supplementary Information. A concise summary is presented below.

(a) Area

Compared with the 1D design, the 2D MTJ-MCA occupies a smaller footprint because the weights are encoded by domain width, eliminating the need for the pair of anomalous Hall (AHE) contacts used in the 1D version. Consequently, the total area is reduced from **133 μm^2 (1D)** to **103 μm^2 (2D)**.

(b) Energy

The 2D design also exhibits significantly lower energy consumption, primarily due to the reduced shifting energy, while the write and read energies remain the same for both architectures. By optimizing material pinning, the required current density for domain-wall motion can be reduced by several orders of magnitude.

In the current estimation, the shifting energy decreases from **1.024 nJ (1D)** to **0.61 pJ (2D)**. This is based on a recent experimental report [1], where optimized pinning in a W/CoFeB/MgO heterostructure resulted in domain-wall displacement energies as low as 0.4 fJ for 18.6 μm motion. Scaling this to our $\sim 200\ \mu\text{m}$ device corresponds to an energy of ~ 4 fJ—roughly five

orders of magnitude lower than the 1D design. To remain conservative, we adopted their worst-case condition, yielding the 0.61 pJ value used in our estimation.

It is worth noting that the material system in Ref. [1]—W/CoFeB/MgO—is nearly identical to ours (Ta/CoFeB/MgO), and Ta can be directly substituted by W, confirming the practical feasibility of this improvement.

(c) Throughput

Throughput in both 1D and 2D architectures is determined by the domain-wall velocity, which remains identical in both cases. As a result, the **throughput is ~248 MHz** for both designs.

Reference

[1] Kumar, Durgesh, et al. "Ultralow energy domain wall device for spin-based neuromorphic computing." *ACS nano* 17.7 (2023): 6261-6274.

10. Any practical use will require integration with CMOS for control, ADC/DAC, and post-processing. Have the authors analyzed the latency and energy of these conversions? How does this impact the claimed system-level advantage?

Reply:

We thank the reviewer for this insightful question. Our response is divided into two sections:

1. Latency and energy associated with CMOS conversions
2. Impact on the system-level advantage

Latency and energy associated with CMOS conversions

To ensure realistic benchmarking, our performance estimation explicitly includes the overhead from all peripheral CMOS circuitry. We apologize that this was not made sufficiently clear in the original manuscript.

Additional details are provided below and summarized schematically in **Fig. R15**, which illustrates the system integration components—drivers, control circuits, ADC, and the electrical–magnetic interfaces. All components were modeled using 28 nm process design kits (PDKs) from TSMC, ensuring accurate estimation of latency, energy, and area, including routing and layout overhead.

- **Reading and Summation Circuitry:** Accounting for approximately 80% of the total area, the summation circuitry handles the addition of the weighted inputs and contributes significantly to the total area and energy metrics.
- **Analog-to-Digital Conversion (ADC) and Domain Writing/Shifting Circuitry:** These components have lower area and energy overhead as they are shared across the entire MCA.

Fig. R15 | The architecture of the evaluated MCA that includes all peripheral components. *This figure is an illustration of the design, the actual circuit layout is significantly larger and is not shown here for brevity.*

More details of **Fig. R15** are provided below.

- **Domain Writing:** The domains are written using a driver that generates a control voltage and drives a current through a metal wire to create a magnetic field, establishing the interface from electrical to magnetic domains.
- **Domain Shifting:** This is achieved via spin-orbit torque (SOT), where a driver generates a control voltage and drives a current through the strip (comprising Heavy Metal/Ferromagnet layers of the convolution device) to shift the magnetic domains within the device, forming the interface from electrical to magnetic domains.
- **Readout:** The anomalous Hall effect (AHE) converts magnetic signals into voltage signals, creating the interface from magnetic to electrical domains.
- **Summation and ADC:** Both summation and ADC operations are fully implemented in the electrical domain. **Following reviewer’s suggestion, we further add an ADC such that the output is not only electrical, but also digitized.**

We adopt the ADC specifications from Reference [1], which has the same 8-bit precision and 28nm technology as our evaluation. The ADC increases the total area by $\sim 200\%$, but has negligible effect on energy and latency overhead as it is shared across the entire MCA and process in real-time. The corresponding details are discussed below.

We evaluate the AHE readout for an 8-bit resolution, 1024-point convolution operation at Gigahertz (GHz) frequencies, which is suitable for most AI and signal processing applications:

- In our experiment, the range of the AHE signal is $0.4mV (+ - 0.2mV)$ for a $0.1mA$ reading current for a single convolution unit (a single Hall device). For a 1024-point convolution, the output voltage range would be $0.4mV \times 1024 = 409.6mV$ and the variation for an 8-bit precision would be $\frac{409.6mV}{2^8} = 1.6mV$, which can be readout by most circuits.
- To achieve the target 8-bit precision, we should compare the voltage range with that of the 28 nm supply. Since our range ($409.6mV$) is similar to the 28 nm V_{DD} ($0.9V$), no additional voltage amplification or range conversion is necessary.

- Based on our device’s operational frequency (248 MHz, as estimated in the manuscript) and output characteristics (8-bit precision), we require an 8-bit ADC capable of operating at > 250 MS/s to match the high-speed requirements. If the ADC throughput is below 250 MS/s, multiple ADC instances would be required for each 1024-point convolution engine. Conversely, if the ADC throughput exceeds this requirement, a single ADC can be shared among multiple MCA devices.

Impact on the system-level advantage

Given the aforementioned specifications, we have added a 28 nm ADC to our benchmarking. The ADC [1] is 0.009 mm² in area, has an 8-bit precision and operates at 10 GS/s, fitting the above requirements and is capable of supporting a large amount of MCA devices. The specification can be found in Reference [1].

The benchmarking performance is updated accordingly: The total energy is increased by 58.9 fJ (ADC conversion energy), the area is increased by $\frac{248 \text{ MHz}}{10 \text{ GHz}} \times 0.009 \text{ mm}^2 = 223.2 \mu\text{m}^2$. The inclusion of the ADC has negligible effect in energy (from 1.03 nJ to 1.0300589 nJ) but results in an approximate 2 × increase in area (from 133 μm² to 356.2 μm²).

We have updated the benchmarking results based on the revisions made here. The updated benchmarking figure is shown in Fig. R16 below, which is also updated in the Main Text Fig. 4b.

Fig. R16 | Updated performance comparison of 1D Hall- and 2D MTJ-MCA against CMOS on the same node. The plot is in log scale. Figure of merit (FOM) is defined as the Throughput (T) divided by the product of Area (A) and Energy Efficiency (E) ($FOM \equiv T/(AE)$).

Reference

[1] Chen, Qian, et al. "17.8 A Single-Channel 10GS/s 8b> 36.4 d8 SNDR Time-Domain ADC Featuring Loop-Unrolled Asynchronous Successive Approximation in 28nm CMOS." 2023 IEEE International Solid-State Circuits Conference (ISSCC). IEEE, 2023.

11. Reliability is critical for deployment. High-current pulses may cause electromigration or heating. Have the authors measured temperature rise during operation? What is the estimated lifetime under continuous cycling? Are there mitigation strategies (e.g., duty cycling, material changes)?

Reply:

We thank the reviewer for this important question regarding device reliability and potential heating effects. Our response is divided into three sections:

1. Additional experiments on temperature rise
2. Additional experiments on lifetime under continuous cycling
3. Mitigation strategies

Additional experiments on temperature rise

The primary sources of heating are high-current pulse for domain motion. To characterize this effect, we performed additional experiments.

First, we measured the R_{xx} - T curve (device longitudinal resistance vs. temperature) for the same devices used in the manuscript. The setup is shown in **Fig. R17a**. A 1 μ A reading current (chosen to minimize self-heating) was applied to the current-shifting channel, and the voltage was measured across the two Hall channels using the standard four-probe method. The resulting R_{xx} - T curve (**Fig. R17b**) shows a resistance of 349.72 Ω at 300 K with a slope of $-0.069 \Omega/K$, which serves as our calibration reference. The temperature rise (ΔT) associated with Joule heating during domain shifting can be obtained from the measured change in longitudinal resistance (ΔR_{xx}) using: $\Delta T = -\frac{1K}{0.069\Omega} \Delta R_{xx}$.

To quantify the heating effect, we applied DC currents ranging from 1 μ A to 1 mA and measured the equilibrium R_{xx} (**Fig. R17c**). For a 1 mA current—the shifting current used in the manuscript—we obtain: $\Delta T = \frac{1K}{0.069\Omega} \Delta R_{xx} \rightarrow \Delta T = -\frac{1K}{0.069\Omega} \times (348.33\Omega - 349.72\Omega) \approx 20 K$. This indicates that the device temperature increases from **300 K to approximately 320 K during 1mA DC current application**.

A temperature rise of $\sim 20 K$ is modest in magnetic device applications and is unlikely to cause irreversible changes in magnetic properties. Larger temperature change is commonly observed in spin-orbit torque and domain-wall motion studies, without performance degradation.

Moreover, in practical operation, the device is driven by pulse with a 50% duty cycle (or lower), which further reduces the effective heating. Therefore, Joule heating is not expected to notably affect the reliability or reproducibility of our device.

Fig. R17 | Heating effect during Shifting operation. (a), Measurement setup. **(b),** R_{xx} vs. T curve. **(c),** R_{xx} as a function of DC current amplitude. Each current is applied for 4 hours, and measurements are performed at room temperature.

Revisions:

This section is added into the Supplementary Fig. S2.

A description is added into the Main Text: From Line 179 to 181.

Additional experiments on lifetime under continuous cycling

The lifetime (endurance) can be divided into two parts, magnetic and non-magnetic.

The magnetic endurance is essentially unlimited, since magnetization reversal itself is inherently non-destructive, it has no wear-out mechanism — as emphasized in fundamental reviews of spintronics and MRAM technology [1–3]. The non-magnetic endurance (mostly electric endurance) — including tunnel barrier reliability, interface degradation, dielectric breakdown, and electrical stress — has been experimentally demonstrated to exceed 10^{14} switching cycles at 90% duty cycle without degradation [4,5].

To experimentally assess our own device stability, we monitored the anomalous Hall effect (AHE) signal under continuous operation. Using the same device from the temperature-rise test, we applied a 1 mA DC current (domain-motion current) and repeatedly measured the AHE loops over five days. More than 1000 AHE loops were recorded (**Fig. R18a**). All curves overlap nearly perfectly, indicating no measurable change in either magnetic (AHE resistance, PMA, DMI, M_s) or electrical (longitudinal resistance, contact resistance) properties.

Assuming a 2 ns operation pulse and 50% duty cycle, this corresponds to a projected endurance of **2.16×10^{14} operation**, which is sufficient for most AI and signal-processing workloads. A longer test duration would likely confirm even higher endurance, but was not pursued due to time constraints. A zoom-in of the coercive-field region (**Fig. R18b**) shows minor fluctuations (~ 0.3 Oe), attributable to expected thermal noise near coercivity.

Fig. R18 | Endurance test for the MCA device. (a), 5-day AHE measurement. Offset in Hz is due to the remanence field of the electro-magnet. **(b),** Zoom-in plot of the red box region of (a), at the coercivity.

Reference

- [1] Chappert, C., Fert, A. & Van Dau, F. The emergence of spin electronics in data storage. *Nature Mater* 6, 813–823 (2007).
- [2] Everspin Technologies Whitepaper. Fast Read/Write • Non-Volatile • Infinite Endurance. (Everspin Technologies, 2016).
- [3] Kent, A., Worledge, D. A new spin on magnetic memories. *Nature Nanotech* 10, 187–191 (2015).
- [4] Kan, J.J., Park, C., Ching, C., Ahn, J., Xue, L., Wang, R., Kontos, A., Liang, S., Bangar, M., Chen, H. and Hassan, S., 2016, December. Systematic validation of 2x nm diameter perpendicular MTJ arrays and MgO barrier for sub-10 nm embedded STT-MRAM with practically unlimited endurance. In 2016 IEEE International Electron Devices Meeting (IEDM) (pp. 27-4). IEEE.
- [5] Kan, J.J., Park, C., Ching, C., Ahn, J., Xie, Y., Pakala, M. and Kang, S.H., 2017. A study on practically unlimited endurance of STT-MRAM. *IEEE Transactions on Electron Devices*, 64(9), pp.3639-3646.

Revisions:

This section is added into the Supplementary Fig. S7.

A description is added into the Main Text: From Line 208 to 214.

Mitigation strategies

In practical systems, several established strategies can be employed to further mitigate potential reliability concerns such as electromigration, localized heating, or dielectric degradation — many of which are already adopted in commercial MRAM technologies.

(a) Duty-cycle optimization and current-pulse engineering

By operating in pulsed mode with low duty cycles (typically <10%), the average thermal load and cumulative electromigration stress are significantly reduced, as demonstrated in STT-MRAM and SOT-MRAM devices. Shorter pulse widths (sub-nanosecond) and optimized rise/fall times further suppress transient temperature spikes.

(b) Material and stack engineering

These can enhance robustness. The use of high-melting-point heavy metals (e.g., W) as spin-orbit layers and thermally stable interfaces (e.g., CoFeB/MgO with optimized annealing) improves electromigration resistance and thermal endurance. These design principles are standard in advanced MRAM fabrication lines, where devices routinely sustain $>10^{13}$ switching cycles without degradation.

(c) Thermal management

This can be further improved through heat-spreading metallic underlayers such as Cu or Ru, which distribute local heat more evenly and improve overall temperature stability.

Together, these strategies — already validated in commercial MRAM and SOT-MRAM platforms — are directly transferrable to our MCA devices, providing a strong foundation for long-term endurance and reliable large-scale integration.

Reviewer #3 (Remarks to the Author):

Dai et al present a spintronic convolutional computing platform that exploits DW motion to perform both computation and data storage, realized through Magnetic Convolutional Accelerator (MCA). They demonstrate a concrete advance toward the vision of a general-purpose computing platform or full CIM realization, enabling applications such as short-time Fourier transforms, convolutional neural networks, and image processing. The demonstrated MCA pioneered in convolutional acceleration while simultaneously integrating memory and computation through spintronic mechanisms. Realization of data processing tasks rely on analogies between domain shifting and sliding-window operations, which offers exceptional scalability, energy efficiency, operation speeds and nonvolatility. I recommend publication if authors can address the following concerns.

Reply:

We sincerely thank the reviewer for the thorough evaluation and constructive feedback, which have greatly improved the rigor, clarity, and overall impact of our manuscript. In response to the reviewer's comments, we have conducted additional experiments, algorithmic validations, and provided further explanations to strengthen both the physical and computational aspects of the work. We have also extensively revised the Main Text and Supplementary Materials to incorporate these updates and enhance the manuscript's clarity and scientific rigor.

A detailed point-by-point response to each comment is provided below:

1. Is the 1D MCA convolution kernel multiplication and Fourier transform implemented through peripheral circuits? How is it realized?

Reply:

We thank the reviewer for this insightful question and apologize for not making this sufficiently clear in the original manuscript. To clarify, we provide additional explanations below. Our response is divided into two sections:

1. 1D MCA convolution kernel multiplication
2. Fourier transform implementation

1D MCA convolution kernel multiplication

The multiplication is directly implemented at the device level through the anomalous Hall effect (AHE). Specifically, this is achieved by assigning the convolution input as domain length L_D and the kernel coefficient as the spacing between the Hall electrode pair W_P , as discussed in the manuscript. *Detailed derivation is provided in (c).*

The multiplication is based on two key physical relationships, illustrated in **Fig. R1**:

(a) $V_{AHE} \propto W_P$

The AHE voltage (V_{AHE}) is proportional to the effective current amplitude within the pair of Hall electrodes. This arises from the current shunting effect: for a fixed read current (I_H), the fraction

of current passing through the Hall electrode pair scales with W_P , the spacing between the electrodes.

(b) $V_{AHE} \propto L_D$

The AHE voltage is also proportional to the net magnetization within the Hall sensing region, which depends on the magnetic domain area. For a fixed device width (W), the magnetized area — and thus the net magnetization — scales linearly with the domain length L_D .

Combine these two, $V_{AHE} \propto W_P \times L_D$, which directly implements the multiplication function at the device level.

Fig. R1 | The convolution device. V_H : AHE voltage. I_H : Reading current. W : Device width. L_D : Domain length. W_P : Spacing between the Hall electrode pair. L_P : Hall electrode length. **Upper picture:** A single unit. **Lower picture:** Different L_D and W_P .

(c) *Detailed derivation*

The MCA device is structured such that the AHE voltage (V_H) can be expressed as:

$$V_H = \alpha' \cdot M_z \cdot I_H \cdot \frac{W_P}{W}$$

Here, α' represents a material-dependent constant, M_z is the net out-of-plane magnetization within the sensing region between a pair of Hall electrodes, I_H is the applied current along the +x direction, W_P is the separation between the two Hall electrodes, and W is the total device width.

The net magnetization M_z is described by:

$$M_z = \left(-2\frac{L_D}{L_P} + 1\right) M_s$$

where M_s denotes the saturation magnetization, L_D is the length of the reversed (-z) magnetic domain, and L_P corresponds to the contact length of the Hall electrodes.

Initially, the Hall sensing area is uniformly magnetized along the +z direction. When a reversed domain of length L_D enters the region, the net M_z decreases linearly with L_D . As α' , M_s , and W are intrinsic material and device constants, they can be consolidated into a single coefficient α for simplicity.

Substituting the expression for M_z into the V_H formula yields:

$$V_H = \alpha I_H (2L_D - L_P) W_P$$

In our implementation, we fix α , L_P , and I_H during operation, enabling a simplified form:

$$V_H = c_1 + c_2 W_P L_D$$

Where c_1 and c_2 are device-specific constants determined by fabrication.

This linear dependence allows the Hall voltage to directly encode the product of domain length L_D and electrode spacing W_P — the two physical quantities mapped to the convolution input and kernel, respectively— enabling on-device multiplication without the need for peripheral circuits.

Fourier transform implementation

The Fourier transform is jointly implemented by the MCA device and computer post-processing. The MCA device performs the physical convolution operation, while the computer performs the final scaling to complete the Fourier transform. *Detailed implementation is provided in (c).*

(a) The MCA device part

The Fourier transform begins with the convolution of the input signal, which is physically executed by the MCA device. The input signal is encoded as the magnetic domain length (L_D). The convolution process consists of three steps: multiplication, shifting, and readout. The **multiplication** mechanism is described in the previous section. The **shifting** is achieved through domain-wall motion, and the **readout** is performed by measuring the anomalous Hall voltage (V_{AHE}), which represents the convolution result.

(b) The computer part

At the current stage, the final scaling of the AHE voltage to complete the Fourier transform is performed digitally. This step is straightforward, as the MCA already outputs an analog signal proportional to the convolution result, and the computer simply applies a multiplicative factor to obtain the Fourier result.

In the future, this operation could be fully migrated into hardware by integrating low-overhead analog peripheral circuits, such as transconductance amplifiers or programmable gain stages [1], directly adjacent to the MCA array. These circuits would perform the necessary scaling in real time, eliminating the need for computer-side post-processing and enabling an entirely on-chip Fourier transform. Importantly, such circuits require only a negligible amount of chip area and

energy, and because the scaling is executed in real time, they do not introduce additional latency or reduce throughput [2]. This integration is consistent with standard mixed-signal design practices in neuromorphic and in-memory computing systems [3, 4].

Reference

- [1] Razavi, B. Design of Analog CMOS Integrated Circuits, 2nd ed.; McGraw-Hill, 2016.
- [2] Baker, R.J., 2019. CMOS: circuit design, layout, and simulation. John Wiley & Sons.
- [3] Sebastian, A., Le Gallo, M., Khaddam-Aljameh, R. et al. Memory devices and applications for in-memory computing. Nat. Nanotechnol. 15, 529–544 (2020).
- [4] Ambrogio, S., Narayanan, P., Tsai, H. et al. Equivalent-accuracy accelerated neural-network training using analogue memory. Nature 558, 60–67 (2018).

(c) Detailed implementation

The Discrete Fourier Transform (DFT) is a mathematical operation that converts a sequence of time-domain samples into a corresponding set of complex-valued frequency-domain coefficients. It can be expressed in a convolution form, enabling its hardware implementation in the MCA device.

The DFT is formally defined as:

$$X_k = \sum_{n=0}^{N-1} x_n \cdot e^{-\frac{i2\pi}{N}kn}$$

Where x_n are the input time-domain samples and X_k are the resulting frequency-domain outputs.

This equation can be algebraically rearranged into a form that resembles a convolution:

$$X_k = e^{-\frac{i\pi k^2}{N}} \sum_{n=0}^{N-1} \left(x_n \cdot e^{-\frac{i\pi n^2}{N}} \right) \left(e^{\frac{i\pi(k-n)^2}{N}} \right)$$

By introducing the substitutions $a_n = x_n \cdot e^{-\frac{i\pi n^2}{N}}$ and $b_n = e^{\frac{i\pi n^2}{N}}$, the expression becomes:

$$X_k = b_k^* \sum_{n=0}^{N-1} a_n b_{k-n}$$

This transformation highlights a convolution structure involving a fixed kernel b_{k-n} . Importantly, "fixed kernel" implies that the coefficients b_{k-n} are independent of the input signal x_n and remain unchanged. In the MCA device, this feature is leveraged by encoding these fixed weights through predefined electrode spacings (W_p) during fabrication.

Since DFT outputs are complex-valued, the convolution must be separated into real and imaginary components. The input signal, pre-multiplied by the complex phase factor $e^{-\frac{i\pi n^2}{N}}$, is first decomposed into real and imaginary parts. Each part is then individually mapped to the domain lengths (L_D) in separate MCA devices. The real and imaginary parts of the convolution kernel b_{k-n} are similarly mapped to dedicated devices. Convolution operations proceed independently within these MCA devices by sequential domain shifting and AHE readout. After completing the convolution across MCA devices (the input real/imaginary parts and the kernel real/imaginary parts), the results are recombined by post-multiplying by b_k^* . The final calculated values X_k reveal the amplitude and phase information across different frequency components, as depicted in **Fig. R2**.

We have revised the Main Text: Line 257 to 259, highlighted in yellow.

These two descriptions are provided in the Main Text.

1. Convolution multiplication: Line 505 to 521.

2. Fourier transform implementation: Line 594 to 619.

Fig. R2 (Extended Data Fig. 8) | Mapping of DFT to the MCA device. Complex number processing is performed by decomposing the input signal into real and imaginary parts, each handled by separate MCA devices. Four MCA devices are used in total to process both components. The combined output is then post-multiplied by b_k^* to yield the Fourier-transformed signal X_k .

2. The accumulated errors during multiple domain wall movements were not addressed. Such errors pose a significant challenge in physical hardware implementations of multiply-accumulate (MAC) operations.

Reply:

We thank the reviewer for this insightful comment. Our response is divided into three sections:

1. Accumulated errors during domain-wall movements
2. Errors in multiply-accumulate (MAC) operations
3. Effect on algorithm accuracy

Accumulated errors during domain-wall movements

The primary source of error arises from domain-wall pinning, which can cause de-synchronization of domain walls during motion and potentially affect MAC precision.

We used MOKE imaging to quantitatively assess domain-wall synchronization during propagation. As illustrated in **Fig. R3a**, a domain is bounded by two domain walls (DWs), and

their spacing—the domain length (L_D)—may fluctuate during motion. Thus, the degree of synchronization can be evaluated by measuring the variation in L_D as the domain moves along the track.

We experimentally prepared two representative cases: a wide domain and a narrow domain. The result is shown in **Figs. R3b & c**. For wide-domain propagation, the measured domain lengths exhibit a standard deviation of **0.15 μm (1.9%)**, and for narrow domains, **0.10 μm (2.1%)**. These results confirm that the accumulated errors are small.

Revisions:

This section is added into the Supplementary Fig. S3, highlighted in yellow.

A description is added into the Main Text: From Line 188 to 191, highlighted in yellow.

Errors in multiply-accumulate (MAC) operations

The anomalous Hall voltage (V_{AHE}) is proportional to the domain length (L_D) and the spacing between pairs of Hall pads (W_P), i.e., $V_{\text{AHE}} \propto L_D \times W_P$, which serves as the basis of the physical MAC operation.

In our experiments, the primary error sources arise from two factors: **(i)** deviations from the ideal domain length (input variation) and **(ii)** deviations from the ideal Hall pad spacing (weight variation). The corresponding variations are shown in Figs. 2g and 2h of the Main Text and

reproduced in **Fig. R4** below. Quantitatively, the average variation in Hall pad spacing is approximately **7%**, while that in domain length is approximately **2%**.

Effect on algorithm accuracy

The observed variations introduce small errors in the MAC function, which could translate into algorithmic deviations. To evaluate their impact, we performed additional experiments as described below.

(a) Input variation

We first examined the effect of input variation by comparing the handwritten-digit dataset processed using the raw experimental data from Fig. 2h (which includes domain-length deviations) with the ideal case. As shown in **Fig. R5**, the images with and without domain-length variation appear visually identical, as the experimental domain-length fluctuation is only $\sim 2\%$. This confirms that the essential image features required for accurate recognition are well preserved.

The corresponding handwritten-digit recognition task, incorporating these nonlinear deviations, achieves an accuracy of **98.94%**, demonstrating that domain-length variation has a negligible effect on the algorithm's performance.

Fig. R5 | Input with and without deviation. Left: MNIST image without incorporating domain variation. **Right:** MNIST image with domain length variation. Both images show a clear digit shape and the difference is negligible.

(b) Weight variation

In the original manuscript, we already accounted for variations in the weights by directly using the raw V_{AHE} readouts from different Hall channels as the weight values.

As discussed in the manuscript, the training process was performed in software, whereas the MCA device is designed for inference with fixed weights. Accordingly, the kernel values derived from the training results are approximated using the nearest experimental V_{AHE} values, as illustrated below:

Ideal convolution kernel (training):

0.1399	-0.289	0.172
0.489	0.251	-0.521
0.2173	-0.299	-0.452

Raw V_{AHE} approximation (experiment):

0.1400	-0.271	0.184
0.490	0.239	-0.524
0.2208	-0.305	-0.478

We then performed inference using both **experimental input** and **weight variations** (raw V_{AHE} values incorporating domain-length and Hall-pad deviations). The resulting recognition accuracy remains high at **98.80%**, confirming that the errors in MAC operations do not significantly impact the algorithm’s performance.

Revisions:

This section is added into the Supplementary Fig. S10.

A description is added into the Main Text: From Line 327 to 330.

3. In a 2D MTJ-based MCA architecture, are the magnetic domain regions in the strips used as convolution kernel coefficients continuously generated in a certain time sequence, or are they fixed within the MTJ region? If they are generated dynamically, how is the simultaneous appearance of magnetic domains in the upper and lower strips within the MTJ region controlled? If they are fixed, how can the corresponding magnetic domain widths be modified for different cases? Does the read current used to measure the MTJ magnetoresistance have any effect on the magnetization direction of the magnetic domains?

Reply:

We thank the reviewer for these thoughtful questions. Our response is divided into three sections:

1. Dynamical generation and control of kernel coefficients
2. Simultaneous control of upper and lower magnetic domains
3. Effect of read current on the magnetization direction

Dynamical generation and control of kernel coefficients

We apologize for not making this point clear in the original manuscript. The kernel coefficients are **dynamically generated and tunable** in a defined time sequence.

As illustrated in **Fig. R6**, the kernel coefficients are encoded in the upper magnetic strip. Similar to the domain generation method used in the Hall-based MCA, the magnetic domains are nucleated and tuned using current-induced Oersted fields generated by a local Oersted channel.

The sequence proceeds as follows: the first domain is generated and its width (corresponding to the coefficient value) is adjusted (**Fig. R6a**); it is then shifted to the first MTJ junction region (**Fig. R6b**). The second domain is subsequently generated (**Fig. R6c**), after which both domains are shifted together into their designated MTJ junctions, and the third domain is generated (**Fig. R6d**). This process continues until all kernel-coefficient domains are deployed. When a new set of kernel coefficients is required, the same procedure can be repeated to redefine the domain pattern.

Thus, the kernel domains are generated sequentially, and they remain **tunable and movable** within the MTJ array.

Revision:

We have added this section into Supplementary Fig. S11.

A description is added into the Main Text: From line 361 to 363.

Fig. R6 | Kernel coefficient domain generation process. **a**, The first domain is created and tune by Oersted field. **b**, The first domain is shifted to the MTJ junction region. **c**, The second domain is generated and tuned. **d**, The first and second domain is shifted to the MTJ junction region. After this, the third domain is generated.

Simultaneous control of upper and lower magnetic domains

The magnetic domains in the upper (kernel) and lower (input) strips can be controlled independently, as illustrated in **Fig. R7**. The upper-strip domains are generated as described above, while the lower-strip domains (input data) are produced using the same sequential domain-generation and shifting approach. Independent Oersted field lines and current channels allow precise temporal and spatial control of both layers, ensuring synchronized alignment of input and kernel domains within each MTJ junction.

Revision:

We have added this section into Supplementary Fig. S12.

A description is added into the Main Text: From line 361 to 363.

Effect of read current on the magnetization direction

In the MTJ structure, the read and write currents share the same vertical conduction path through the junction. This current can, in principle, influence the magnetization direction of the free layer via the spin-transfer torque (STT) mechanism, which is the fundamental operation principle of STT-MRAM.

However, the critical current density required for STT-induced switching is approximately 3.9×10^{10} A/m², as reported in the landmark study on perpendicular MTJs [1]. In practice, the read current is chosen to be sufficiently large to produce a measurable voltage signal but at least two orders of magnitude smaller than the critical switching current to prevent any disturbance of the magnetization state.

To verify this experimentally, we used our MTJ stack (**Fig. R8a**) to fabricate MTJ-MCA devices (**Fig. R8b**). Additional details on the MTJ-MCA structure are provided in **Comment 4**. A read voltage of 20 mV typically provides a clear TMR signal. To examine whether higher read currents influence the magnetization, we applied read voltages of ± 200 mV, corresponding to currents of approximately ± 3.1 mA (current densities of $\pm 7.7 \times 10^6$ A/m²). As shown in **Fig. R8c**, no noticeable change is observed in the TMR loops for these two cases.

Although \pm STT could, in principle, act as an effective $\pm z$ -directional field and shift the TMR loops, the nearly perfect overlap of the two loops in **Fig. R8c** confirms that the read current does not influence the domain magnetization.

In conclusion, the applied read current is sufficient to resolve the TMR signal while remaining well below the threshold for disturbing the magnetization direction of the magnetic domains.

Fig. R8 | Reading current test. (a), MTJ material stack. (b), MTJ-MCA device. (c), TMR loop measured by high reading voltages (currents).

Reference

- Ikeda, S., Miura, K., Yamamoto, H. et al. A perpendicular-anisotropy CoFeB–MgO magnetic tunnel junction. *Nature Mater* 9, 721–724 (2010).

4. The fixed-weight approach is only suitable for inference, not for the computationally intensive and diverse training phase. Additionally, the 2D architecture lacks experimental validation.

Reply:

We thank the reviewer for this valuable comment. We fully agree that while the fixed-weight MCA device provides substantial advantages for inference, it is not suitable for the training stage, which requires tunable weights. Therefore, the 2D MTJ-MCA architecture with programmable weights is essential for enabling on-chip training.

To address this, we have **experimentally validated the 2D MTJ-MCA architecture** and demonstrated tunable-weight operation suitable for training. The results are presented in two parts:

1. Fabrication of MTJ-MCA devices on an industry-grade full-MTJ material stack (8-inch wafer)
2. Experimental demonstration of Read, Write, Shift, and Tunable Weight operations

Fabrication of MTJ-MCA devices on an industry-grade full-MTJ material stack (8-inch wafer)

In collaboration with a semiconductor foundry, we developed an industry-grade full-MTJ material stack on an 8-inch wafer. The wafer photograph and corresponding material stack are shown in **Fig. R9a** and **Fig. R9b**, respectively. Using this stack, we fabricated test MTJ-MCA devices, as shown in the optical micrograph in **Fig. R9c**. Each device consists of a magnetic strip (the Shifting Channel), a MTJ readout region (the Reading Junction, highlighted by the red dashed box), and two Oersted-field writing lines (the Writing Channel). The schematic on the right of **Fig. R9c** illustrates the detailed structure within the red dashed box.

Fig. R9 | Industry-grade full-MTJ material stack 8-inch wafer and MTJ-MCA device. (a), A photo of the 8-inch wafer. **(b),** Magnetic tunneling junction material stack. The layer thicknesses are not provided for confidential reasons. **(c),** MTJ-MCA device. The structure inside the red dashed box is illustrated in the right schematic. Red: $-M_z$ domain. Blue: $+M_z$

domain.

Experimental demonstration of Read, Write, Shift, and Tunable Weight operations

(a) Write:

Writing is performed using the Oersted field, same method as the Hall-MCA device. Representative MOKE snapshots are shown in **Fig. R10a**.

(b) Read:

Reading is performed using tunneling magnetoresistance (TMR), which, analogous to the anomalous Hall effect, is proportional to the net M_z . The measured TMR loop of the MTJ-MCA device is shown in **Fig. R10b**. The zoomed-in region (right panel) reveals distinct resistance steps, each corresponding to a convolution input value, with several hundred inputs achieved across the full loop.

(c) Shift:

Shifting is realized through spin-orbit torque (SOT)-driven domain-wall motion, employing the same mechanism as in the Hall-MCA. MOKE snapshots are provided in **Fig. R10c**.

Fig. R10 | Demonstration of Read, Write, and Shift and tunable weight in the MTJ-MCA device. (a), MOKE images of the Domain Writing process. Green shade: Oersted field channel. Yellow arrow: Applied current direction. A current is applied, and a +z domain is created, location mark by the blue arrow. **(b),** TMR loop. Each step corresponds to a convolution input.

Right figure: Zoom-in plot of the red box region. **(c)**, MOKE images of domain wall motion. Yellow shade: Reading junction. t1 to t6 represent the time stamps. Black arrow denotes the motion direction. **d(i)**, TMR loops by applying different junction voltages. Weight is obtained by $(\frac{R_{ap}-R_p}{R_p})$. **d(ii)**, Plot summarizing the measured R_p and R_{ap} at different applied voltages. Inset: weight values $(\frac{R_{ap}-R_p}{R_p})$. Currently, 15 different weights are obtained.

(d) Tunable weights:

In the Hall-MCA, the convolution weights are fixed by the geometric width between pairs of Hall pads. In contrast, the MTJ-MCA enables tunable weights, achieved by modulating the TMR ratio, defined as $\frac{R_{ap}-R_p}{R_p}$, where R_p (R_{ap}) denote the resistances corresponding to parallel and antiparallel alignments of the top and bottom CoFeB layers, respectively.

By applying different junction voltages, the TMR loops can be rescaled. As shown in **Fig. R10d(i)**, each junction voltage produces a distinct TMR loop that differs only in its TMR ratio. In other words, these loops can be transformed into one another by a multiplicative factor—corresponding to the definition of a weight.

As discussed in the **(b) Read** section, each point on the TMR loop represents a convolution input value. By adjusting the TMR ratio, the multiplicative factor (weight) of each input can thus be tuned. For instance, Point 1 at 50 mV transforms into Point 2 at 500 mV, effectively multiplying the input by a scaling factor (weight). Measurements under multiple junction voltages yielded a wide range of tunable weights, summarized in **Fig. R10d(ii)**.

The underlying mechanism is that increasing the junction voltage (and thus the electric field) allows deeper, un-spin-polarized electronic states to contribute to tunneling, thereby reducing the TMR ratio. We note that this voltage-controlled method is simpler than the originally proposed scheme in the Main Text, where the weights are tuned by varying the magnetization of the top CoFeB layer. However, the voltage-controlled range is limited, as higher voltages may induce barrier breakdown. Therefore, the magnetization-based approach remains the preferred and more scalable method for future implementations.

In summary, we have experimentally validated the 2D MTJ-MCA architecture.

5. In line 104 and 105, the authors depict ‘I_H is the applied current flowing along the +x axis, L_D is the domain length aligned along the -z direction’. It seems not the case as illustrated in Fig. 1b.

Reply:

We thank the reviewer for pointing out this issue and apologize for the confusion. The clarification is as follows:

In Fig. 1b, the schematic shows I_H appearing to flow along the -x direction; however, the current path forms a loop, and within the device channel, I_H indeed flows along the +x direction.

The parameter L_D represents the domain length measured along the x-axis, with the domain magnetization oriented along the $-z$ direction.

We have revised the text to read:

“ I_H is the applied current flowing along the $+x$ axis of the device, and L_D is the domain length in the x direction with the domain magnetization aligned along the $-z$ direction.”

This correction has been updated in Lines 107 to 109 of the Main Text.

Fig. R11 | Fig. 1b in the manuscript.

6. In Equation 1, the length of the domain wall is not taken into consideration. Will the DW length influence the final convolutional result?

Reply:

We thank the reviewer for this insightful question. The length of the domain wall (DW) does not influence the final convolution result. The reasons are as follows:

The convolution operation is based on the anomalous Hall effect (AHE). As described in the manuscript, Equation (1) expresses the relationship between the domain length (L_D) and the AHE voltage:

$$V_{AHE} = \alpha I_H (2L_D - L_P) W_P \quad (1)$$

As the reviewer noted, the DW length is not included in the equation. This is because the AHE signal is proportional to the net out-of-plane magnetization (M_z).

According to Reference [1], the anomalous Hall resistivity (ρ_{AHE}) is related to M_z by

$$\rho_{AHE} = R_s M_z$$

Where R_s is a material-dependent constant. Therefore, a zero M_z gives rise to a zero AHE signal.

Within the DW, the local magnetization continuously rotates from $+z$ to $-z$ (or vice versa), resulting in a net M_z of zero, as illustrated in **Fig. R12**. Consequently, the DW does not contribute to the AHE signal and therefore has no influence on the convolution results.

[FIGURE REDACTED]

Fig. R12 | Domain wall magnetic moment. Figure adapted from [2]

Reference

[1] Nagaosa, N., Sinova, J., Onoda, S., MacDonald, A.H. and Ong, N.P., 2010. Anomalous hall effect. *Reviews of modern physics*, 82(2), pp.1539-1592.

[2] Kézsmárki, I., Bordács, S., Milde, P. et al. Néel-type skyrmion lattice with confined orientation in the polar magnetic semiconductor GaV4S8. *Nature Mater* 14, 1116–1122 (2015).

7. Since the domain motion is driven by SOT of the underlying heavy metal, will the external magnetic field used for AHE signal read out affect the domain motion?

Reply:

We thank the reviewer for this important question and apologize for any confusion caused by our earlier description. The experimental procedure consists of three sequential stages:

- 1. Domain creation:** Magnetic domains are generated using the current-induced Oersted field.
- 2. Domain shifting:** The domains are then displaced by spin-orbit torque (SOT).
- 3. AHE signal readout:** The anomalous Hall voltage is measured after the domain displacement.

Importantly, no external magnetic field is applied during either the domain motion or AHE readout stages.

The AHE signal measurement does not require a magnetic field, as it is directly proportional to the net out-of-plane magnetization according to $\rho_{AHE} = R_s M_z$ (discussed in the last comment).

8. In Fig. 2f, the y-intercept of V_{AHE} versus W_P seems not zero (about 0.5 mV). What is the reason and can it be adjusted or eliminated?

Reply:

We thank the reviewer for this careful observation. The non-zero y-intercept of V_{AHE} versus W_P arises from slight misalignment of the Hall pads during fabrication. We clarify the origin and describe possible mitigation methods below. Our response is divided into two sections:

1. Misalignment and non-zero y-intercept of V_{AHE}
2. Mitigation methods

Misalignment and non-zero v -intercept of V_{AHE}

As shown in **Fig. R13a**, during readout, a longitudinal current is applied and the transverse voltage is measured. In an ideally aligned configuration, only the transverse electric field (E_y) from the anomalous Hall effect contributes to the signal.

However, if the Hall pads are slightly misaligned, the voltmeter also detects a small component of the longitudinal electric field (E_x) generated by the read current. This mixing introduces an offset in the measured V_{AHE} . An example is shown in **Fig. R13b**, where the offset exceeds 2 mV. In the ideal case, the V_{AHE} loop would be centered at $V=0$.

Fig. R13 | Aligned and mis-aligned Hall pads and offset of V_{AHE} . (a), AHE measurement. A longitudinal current is applied and the transverse voltage is measured. Only E_y is measured for aligned Hall pads. An additional E_x is measured for the mis-aligned one, which creates an offset of in the V_{AHE} . (b), An example of the V_{AHE} loop from the mis-aligned device.

Mitigation methods

To mitigate or eliminate this offset, we employed higher-resolution lithography for defining the Hall pad pattern, as shown in **Fig. R14**, where the offset is nearly removed. Further improvement can be achieved through foundry-level fabrication, where advanced stepper lithography and automated overlay correction routinely achieve alignment tolerances below 5 nm, effectively eliminating such offsets.

Fig. R14 | AHE measurement of device with improved fabrication. The alignment is improved and the offset is nearly zero.

Even without fabrication upgrades, this offset can be electronically compensated using a differential amplifier or offset-cancellation circuit. For example, if 1000 Hall pad pairs each exhibit an average offset of ~ 0.5 mV, the total accumulated offset would be ~ 0.5 V. A simple differential circuit can subtract this voltage in real time. Such offset-cancellation techniques are standard in Hall sensors and magnetic memory readout systems [1, 2], requiring only minimal power (microwatts) and transistor count, and they operate well within the bandwidth of magnetic device readout [3].

Reference

- [1] Ramsden, E., 2011. Hall-effect sensors: theory and application. Elsevier.
- [2] Popovic, R.S., 2003. Hall effect devices. CRC Press.
- [3] Carusone, T.C., Johns, D.A. and Martin, K.W., 2011. Analog integrated circuit design. John Wiley & Sons.

9. In Fig. 2h, the legend of the picture is incomplete with the absence of corresponding symbol or color.

Reply:

We thank the reviewer for the careful review and for helping improve the quality of our manuscript.

The missing legend in Fig. 2h has been added. The revised figure is provided below as **Fig. R15** for the reviewer's reference and has also been updated in the Main Text.

Fig. R15 | Revised Fig. 2h to include the complete legend. All colors are labeled.

10. In Fig. 3c, the distinction of fabricated W_P is not evident. Will it influence the mapping to convolutional kernel?

Reply:

We thank the reviewer for the careful examination of our manuscript. The distinction in the fabricated W_P values is sufficient for accurate mapping to the convolution kernel.

In Fig. 3c, the device is used to perform the short-time Fourier transform (STFT). The designed W_P sequence for this device is 6–8–6–8–6–8–6 μm , as labeled in **Fig. R16a(i)**. The corresponding measured V_{AHE} values are shown in **Fig. R16a(ii)**. The $V_{\text{AHE}} - W_P$ dependence confirms the intended mapping between W_P and the convolution kernel coefficients.

Although small variations exist among the measured V_{AHE} values, this level of precision is sufficient for algorithmic implementation. As demonstrated in Fig. 3 of the manuscript, the Fourier transform was performed using the raw data from **Fig. R16a(ii)**, yielding accurate reconstruction of the Fourier amplitudes (**Fig. R16b(i)**; corresponding to Fig. 3c(iv) in the Main Text). A zoomed-in view of the boxed region [**Fig. R16b(ii)**] shows only small fluctuations due to fabrication variations in W_P , which do not affect the overall transform accuracy.

Revision:

We have added this section into Supplementary Fig. S8.

A description is added into the Main Text: From Line 261 to 263.

11. In Fig. 4a(iii), the red, blue and purple squares can coexist simultaneously in the crossing area or is just a phenological illustration of all possible cases?

Reply:

We thank the reviewer for the careful review and apologize for not making this point sufficiently clear in the original manuscript. The red, blue, and purple squares in Fig. 4a(iii) can indeed coexist simultaneously in the crossing area. To clarify this, we have redrawn Fig. 4a(iii) as **Fig. R17**, following the same color code and the definitions of L_{DX} , L_{DY} , and W .

The top and bottom magnetic strips overlap in the junction region (crossing area), as illustrated in **Fig. R17a(i)**. Within this region, each magnetic layer contains two domains, represented by red and blue rectangles in **Fig. R17a(ii)**. As shown in **Fig. R17b**, four possible domain configurations can exist in the overlapping area:

1. Region ①: Top:-z, Bottom:-z

2. Region ②: Top:+z, Bottom:-z
3. Region ③: Top:+z, Bottom:+z
4. Region ④: Top:-z, Bottom:+z

Here, the **red** squares correspond to overlapping $(-z, -z)$ domains (region ①), the **blue** squares to overlapping $(+z, +z)$ domains (region ③), and the **purple** squares to $(+z, -z)$ or $(-z, +z)$ domains (region ② ④).

In summary, the red, blue, and purple regions represent the **coexisting domain configurations** of the top and bottom layers within the crossing area. They collectively describe one possible magnetic state; different configurations can occur if the domain positions or lengths (L_{DX} , L_{DY}) are varied—for example, when the top domain becomes entirely $-z$ in the overlapping region.

Revision:

We have added this section into Supplementary Fig. S13.

A description is added into the Main Text: From line 365 to 366.

Fig. R17 | Schematic of the MTJ-MCA. (ai), The MTJ-MCA schematic. Blue: +z domain. Red: -z domain. Yellow: Junction region (crossing area). Strip width: W . Top -z domain length: $L_{D,X}$. Bottom -z domain length: $L_{D,Y}$. **(a ii)**, Top and bottom domain configuration at the crossing area. **(b)**, Detailed view of the crossing area. It can be divided into four regions. Red square (region ①): Top:-z Bottom:-z; Purple square (region ②): Top:+z Bottom:-z; Blue square (region ③): Top:+z Bottom:+z; Purple square (region ④): Top:-z Bottom:+z. The corresponding 3D view are shown.

12. As claimed in line 362 and 363, energy consumption per operation in the 1D Hall-based MCA is primarily due to the large current densities required to drive high-speed domain wall

motion. Will adoption of magnetic solitons like skyrmions or driving method like spin wave in insulators facilitate reduction of energy consumption?

Reply:

We thank the reviewer for this insightful question. Our response is divided into two sections:

1. Adoption of magnetic solitons such as skyrmions
2. Adoption of driving methods such as spin waves in insulators

Adoption of magnetic solitons such as skyrmions

Magnetic solitons, such as skyrmions, are typically much smaller than domain walls—down to a few nanometers in diameter, as experimentally demonstrated in Reference [1]. Their topological spin texture also makes them significantly less sensitive to pinning, allowing them to bypass defects [2,3] or move smoothly along grain boundaries [4,5].

As a result, skyrmions can be driven with minimum current densities of $\sim 10^2$ A cm⁻², which are 3–6 orders of magnitude lower than those required for domain-wall motion (10^5 – 10^8 A cm⁻²) [6–10].

In summary, the use of magnetic solitons such as skyrmions could substantially reduce energy consumption due to their smaller size, lower pinning sensitivity, and dramatically lower drive current density.

Adoption of driving method such as spin wave in insulators

We consider two cases:

(a) Ferromagnetic (FM) insulator

In a purely FM insulator, domain walls or other spin textures can be driven by spin waves. However, FM insulators generally lack a strong electrical readout mechanism compared to metallic ferromagnets such as CoFeB, which exhibit pronounced anomalous Hall and tunneling magnetoresistance signals. This limits the ability to electrically detect or read out the resulting states.

(b) FM insulator as layer below CoFeB

Alternatively, a hybrid stack such as FM insulator/CoFeB/MgO (e.g., YIG or TmIG as the insulator) can be used. In this case, spin waves generated in the insulator can transfer angular momentum across the interface and drive domain walls or skyrmions in the CoFeB layer. Since magnons carry spin angular momentum of \hbar , compared to $\hbar/2$ for electrons, this mechanism is intrinsically more efficient.

Experimentally, magnon-driven domain-wall motion has been demonstrated in Bi-doped YIG, with an upper-bound energy consumption of ≈ 4 pJ to move a domain wall by 15 μm [11]. In comparison, our current device requires ≈ 143 pJ for the same displacement, indicating that spin-wave-based driving can indeed reduce energy consumption.

However, this high efficiency is achieved when the spin wave and domain wall coexist in the same FM insulator. In a bilayer system (FM insulator/CoFeB), the spin wave must transmit

through the interface to exert torque on the CoFeB, reducing overall efficiency. To our knowledge, such interfacial magnon-driven motion has not yet been experimentally quantified, though it is reasonable to expect improved energy efficiency relative to current-driven domain-wall motion.

In summary, both approaches—using magnetic solitons such as skyrmions and adopting spin-wave-based driving in magnetic insulators—offer promising routes to reduce the energy consumption required for high-speed domain-wall motion in future MCA devices.

Reference

- [1] Romming, N., Kubetzka, A., Hanneken, C., von Bergmann, K. and Wiesendanger, R., 2015. Field-dependent size and shape of single magnetic skyrmions. *Physical review letters*, 114(17), p.177203.
- [2] C. Reichhardt, D. Ray, and C. J. Olson Reichhardt, Collective Transport Properties of Driven Skyrmions with Random Disorder, *Phys. Rev. Lett.* 114, 217202 (2015).
- [3] J. Iwasaki, M. Mochizuki, and N. Nagaosa, Current induced skyrmion dynamics in constricted geometries, *Nat. Nanotechnol.* 8, 742 (2013).
- [4] A. Salimath, A. Abbout, A. Brataas, and A. Manchon, Current-driven skyrmion depinning in magnetic granular films, *Phys. Rev. B* 99, 104416 (2019).
- [5] X. Gong, H. Y. Yuan, and X. R. Wang, Current-driven skyrmion motion in granular films, *Phys. Rev. B* 101, 064421 (2020).
- [6] Jonietz, F. et al. Spin transfer torques in MnSi at ultralow current densities. *Science* 330, 1648–1651 (2010).
- [7] Yu, X., Kanazawa, N., Zhang, W. et al. Skyrmion flow near room temperature in an ultralow current density. *Nat Commun* 3, 988 (2012).
- [8] Yamanouchi, M., Chiba, D., Matsukura, F. & Ohno, H. Current-induced domain-wall switching in a ferromagnetic semiconductor structure. *Nature* 428, 539–542 (2004).
- [9] Thomas, L., Moriya, R., Rettner, C. & Parkin, S. Dynamics of magnetic domain walls under their own inertia. *Science* 330, 1810–1813 (2010).
- [10] Togawa, Y. et al. Current-excited magnetization dynamics in narrow ferromagnetic wires. *J. Appl. Phys.* 45, L683–L685 (2006).
- [11] Fan, Y., Gross, M.J., Fakhrul, T. et al. Coherent magnon-induced domain-wall motion in a magnetic insulator channel. *Nat. Nanotechnol.* 18, 1000–1004 (2023).

Reviewer #1 (Remarks to the Author):

Overall, the authors have carefully addressed my major concerns and strengthened the work with new data, making the revised manuscript more rigorous. The work highlights the potential of MCA for convolutional computing, especially in energy and area efficiency, contributing to emerging computing paradigms. While most issues are resolved, I have a few minor questions for further clarification or refinement:

1. While the authors correctly note that advanced material systems (e.g., ferrimagnets, SOC materials) could enhance performance, the challenges of integrating these into a functional MCA platform, such as interface engineering and read/write compatibility, are not clearly addressed. A brief acknowledgement of these integration challenges as a direction for future work would strengthen the discussion and enhance its rigor.
2. The conclusion section primarily highlights the benefits and potential research directions of the MCA, but provides limited discussion on its inherent limitations (e.g., energy consumption, constraints of the 1D design, stochastic DW motion). Could a short discussion be added to the discussion or conclusion addressing the current or potential limitations and the challenges that need to be overcome for using MCA as a compute-in-memory device?
3. Some figure captions (for example, Fig. 1c) appear quite lengthy and largely restate content already covered in the main text. Could these captions be rephrased or more concise?

Reply:

We sincerely thank the reviewer for the positive assessment and the constructive suggestions. We have carefully addressed each point in the detailed responses below and revised the manuscript accordingly.

Detailed point-by-point responses are provided below:

1. While the authors correctly note that advanced material systems (e.g., ferrimagnets, SOC materials) could enhance performance, the challenges of integrating these into a functional MCA platform, such as interface engineering and read/write compatibility, are not clearly addressed. A brief acknowledgement of these integration challenges as a direction for future work would strengthen the discussion and enhance its rigor.
2. The conclusion section primarily highlights the benefits and potential research directions of the MCA, but provides limited discussion on its inherent limitations (e.g., energy consumption, constraints of the 1D design, stochastic DW motion). Could a short discussion be added to the discussion or conclusion addressing the current or potential limitations and the challenges that need to be overcome for using MCA as a compute-in-memory device?

Reply:

We thank the reviewer for these valuable suggestions. We have substantially revised the **Discussion and Conclusion** section to address both the material integration challenges and the limitations and challenges of the MCA platform. For clarity, the added discussion is shown in red font below.

Discussion and Conclusion

In conclusion, we demonstrate a hardware platform capable of performing convolution through sequential magnetic DW motion, with applications spanning deep learning, computer vision, and signal and image processing. Benchmarking results reveal performance enhancements of three to five orders of magnitude over 28 nm CMOS technology. By directly mapping magnetic-domain dynamics onto convolutional operations, the MCA unifies data storage (via magnetic domains) and computation (via DW motion and Hall-voltage readout) within a nonvolatile physical framework, establishing a spintronic compute-in-memory architecture. The experimental realization of convolution—a core building block of modern machine learning and signal processing—thus represents a concrete step toward general-purpose computing platforms beyond Boolean logic.

Despite these advancements, the present MCA prototype exhibits several limitations that define key directions for future development. Pinning-induced variations lead to relatively high energy consumption and occasional stochasticity in DW motion, underscoring the need for improved material uniformity and engineered pinning profiles. The current one-dimensional, fixed-weight design restricts kernel programmability; implementing a two-dimensional array with electrically tunable MTJ weights will be essential for general-purpose compute-in-memory functions. As device density increases, additional system-level challenges—including inter-track crosstalk, modest Hall-signal margins, TMR variability across MTJ arrays, and fabrication-yield constraints—must also be addressed to achieve large-scale integration.

Looking ahead, two natural directions emerge. First, the MCA architecture will need to scale in both dimension and functionality to accommodate larger training datasets and more complex workloads, such as those required by models like ChatGPT³⁴. Extending the design into a two-dimensional MTJ-MCA array with programmable weights provides a logical pathway toward versatile convolutional kernels and on-chip learning, and the MCA's simple DW-based racetrack geometry³⁵ facilitates integration atop existing CMOS platforms.

Second, advanced material systems offer promising opportunities to enhance performance. Strong-DMI multilayers²¹ and ferrimagnets²⁴ could enable sub-10 nm domain scaling²¹ and operational speeds approaching ~ 1 THz²⁴, while optimization of pinning landscapes may further reduce DW driving energies to the attojoule regime²². However, realizing these benefits requires overcoming nontrivial materials-integration challenges. In ferrimagnets, the angular-momentum compensation point is highly sensitive to composition and temperature²⁴, and deviations can suppress DW velocity and degrade writing efficiency. In strong-DMI multilayers²¹, large DMI may stabilize skyrmion textures^{4,5,6} rather than the alternating-domain configuration required for MCA operation, necessitating careful tuning of anisotropy, DMI, exchange, and dipolar energies. Moreover, compatibility with MTJ stacks and CMOS back-end-of-line processes imposes stringent constraints on interface engineering, thermal management, and interconnect routing to preserve high TMR readout and robust sensing. Addressing these material, device, and integration challenges will be essential for translating the performance potential of advanced stacks into practical MCA hardware.

Together, these results establish the MCA as a promising pathway toward scalable, energy-efficient spintronic computing, offering a solid foundation for future advances in compute-in-memory hardware.

3. Some figure captions (for example, Fig. 1c) appear quite lengthy and largely restate content already covered in the main text. Could these captions be rephrased or more concise?

Reply:

We thank the reviewer for this valuable suggestion. We have refined all figure captions to improve conciseness and clarity, and the revised captions are provided below for the reviewer's reference.

Fig. 1 | Operating principles of the MCA. (a) MCA architecture comprising serially connected magnetic unit cells. **(b)** Structure of a single unit cell. Red and blue rectangles denote domains with $-z$ and $+z$ magnetization. Gold strips and pads indicate AHE readout electrodes. The domain length is L_D , the electrode contact length is L_P , and the spacing between Hall-electrode pairs is W_P . During readout, a current I_H flows along the conduit and the transverse Hall voltage V_H is measured. **(c)** Schematic of MCA-based convolution. The input domains (top) and kernel coefficients (second row) are represented by discrete values of L_D and W_P . The device-level cartoon (third row) illustrates stationary electrodes and sequential domain shifting. The resulting Hall-voltage sequence (bottom) corresponds to the convolution output. **(d)** DW shifting driven by SOT. A charge current j generates a transverse spin current j_s , which applies a torque on the DWs and translates the domains along the track.

Fig. 2 | Experimental realization of the MCA. (a) Schematic of domain nucleation using current-induced Oersted fields. (b) Schematic of SOT-driven DW shifting. (c) Optical micrograph of a fabricated MCA device. (d) Experimental demonstration of domain nucleation and size control. (i) Setup for generating domains. (ii) Creation of a $32\mu\text{m} +z$ domain using a 200 mA, 1 ms pulse. (iii) Domain-length tuning: pulses of -80 mA , -90 mA , and -100 mA (1 ms) yield $L_D=14$, 11, and 6 μm , respectively. Scale bar: 20 μm . (e) Experimental demonstration of sequential domain shifting. (i) Setup for applying SOT-driven shifting pulses. (ii) Initial domain position. (iii–vi) Stepwise 20 μm displacement per pulse, shifting domains into the Hall-channels. (vii) Continued pulses translate the full domain sequence across the Hall-channels. Scale bar: 20 μm . (f) AHE-based convolution granularity versus W_p at fixed $L_D=14\mu\text{m}$. Shaded color bands indicate error bars. Insets: device region with varying W_p and measurement setup. Scale bar: 14 μm . (g) AHE-based convolution granularity versus L_D at fixed $W_p=8\mu\text{m}$. Insets: measurement setup and snapshots of domains with different L_D . Scale bar: 20 μm . (h) Linear dependence of V_{AHE} on L_D for various electrode spacings W_p (10, 8, 7, 6, and 3 μm).

Fig. 3 | Algorithm implementation on the MCA. (a) Conceptual illustration of the STFT. The input signal is segmented and Fourier-transformed. (b) Schematic of 4-point STFT using a 7-Hall-pad MCA device. Input values a_n map to L_D , and kernel coefficients b_{k-n} to W_P . Sequential domain shifting and AHE readout yield Fourier components X_n . (c) Magnetic-domain snapshot as domains shift across the 7-Hall-pad device. Scale bar: 14 μm . (d) Experimental STFT demonstration. (i) Initial segment of $f(t)$. (ii) Final segment of $g(t)$. (iii) Combined input $f(t) + g(t)$. (iv) MCA-generated STFT output showing the temporal evolution of both components. (e) CNN flow diagram for MNIST handwritten digit recognition. (f) Implementation of a 3×3 convolutional kernel using three MCA devices, each encoding a 1×3 kernel via W_P configurations (grey dots). Pixel intensities map to L_D . Scale bar: 20 μm . (g) Confusion matrix for the MNIST task, showing 98% average accuracy. (h) Left: 256×256 greyscale ‘Cameraman’ image. Right: Intensity profile along a line cut; x-axis denotes pixel position. (i) Experimental setup for edge detection using the kernel $[1, 0, -1]$ encoded via W_P , with pixel intensities mapped to L_D . Scale bar: 20 μm . (j) Edge-detected image produced using the MCA.

Fig. 4 | Two-dimensional MTJ-based MCA architecture and benchmarking against CMOS. (a) Schematic of the 2D MTJ-based MCA (MTJ-MCA). **(i)** Conceptual design using crossing arrays of MTJs. Grey arrows indicate domain-shifting currents; yellow arrows denote vertical readout through the MTJs. **(ii)** Schematic of a single MTJ junction. **(iii)** Top view of an MTJ junction and overlapping magnetic strips. Red and blue regions represent $-z$ and $+z$ magnetizations, with domain lengths $L_{D,Y}$ and $L_{D,X}$ along the y - and x -directed strips and junction width W . Overlap regions highlight different domain combinations. **(b)** Benchmarking comparison between the 1D Hall-based MCA, the 2D MTJ-MCA, and CMOS (28 nm). The plot uses a logarithmic scale. The figure of merit is defined as $\text{FOM}=\text{T}/(\text{AE})$.

Reviewer #2 (Remarks to the Author):

Authors have successfully addressed the prior review comments. I am glad with the current version of the manuscript and recommend acceptance.

Reply:

We sincerely thank the reviewer for the positive assessment and for recommending acceptance. We appreciate your time and valuable feedback throughout the review process.

Reviewer #3 (Remarks to the Author):

The authors have adequately addressed the points raised by the referees. I therefore recommend acceptance of the manuscript in its current form.

Reply:

We sincerely thank the reviewer for the positive assessment and for recommending acceptance. We appreciate your time and valuable feedback throughout the review process.